# Diatomic iron nanozyme with lipoxidase-like activity for efficient inactivation of enveloped virus

Beibei Li[1,2,3,8], Ruonan Ma[4,8], Lei Chen[4,5,8], Caiyu Zhou[4,8], Yu-Xiao Zhang [1], Xiaonan Wang[4], Helai Huang[1], Qikun Hu [1], Xiaobo Zheng[2], Jiarui Yang [2], Mengjuan Shao[6], Pengfei Hao[7], Yanfen Wu[1], Yizhen Che[1], Chang Li [7], Tao Qin [6], Lizeng Gao [4] ✉, Zhiqiang Niu [1] ✉ & Yadong Li[2]

Enveloped viruses encased within a lipid bilayer membrane are highly contagious and can cause many infectious diseases like influenza and COVID-19, thus calling for effective prevention and inactivation strategies. Here, we develop a diatomic iron nanozyme with lipoxidase-like (LOX-like) activity for the inactivation of enveloped virus. The diatomic iron sites can destruct the viral envelope via lipid peroxidation, thus displaying non-specific virucidal property. In contrast, natural LOX exhibits low antiviral performance, manifesting the advantage of nanozyme over the natural enzyme. Theoretical studies suggest that the Fe-O-Fe motif can match well the energy levels of $Fe_2$ minority β-spin d orbitals and pentadiene moiety π* orbitals, and thus significantly lower the activation barrier of *cis,cis*-1,4-pentadiene moiety in the vesicle membrane. We showcase that the diatomic iron nanozyme can be incorporated into air purifier to disinfect airborne flu virus. The present strategy promises a future application in comprehensive biosecurity control.

Enveloped viruses are composed of external lipid membranes to protect the genetic material in their life cycle. Many infectious diseases are caused by enveloped viruses, such as Ebola virus, influenza virus, Dengue fever virus, Zika virus, and SARS-CoV-2[1–7]. Taking influenza A viruses (IAVs) for example, they have caused four worldwide influenza pandemics since the last century[4,8]. Various vaccines and antiviral drugs have been developed for the prevention and treatment of IAVs[8,9]. However, antigenic shift and drift, drug resistance, and the diversity of influenza subtypes can compromise the effectiveness of these strategies[9–11]. Therefore, it is imperative to develop

comprehensive biosecurity control measures such as environmental disinfection and interruption of virus transmission.

Nanozymes are synthetic nanomaterials with enzyme-like catalytic properties[12–14]. Compared with natural enzymes, nanozymes have the advantages of high stability, low cost, and long-term storage, which endow them with wide applications in biotherapy, biosensing, biocatalysis, antiviral treatment, and environmental remediation[15–20]. Single atom catalyst (SAC) is emerging as a new class of nanozymes[21–23]. SAC consists of atomically dispersed metal sites supported on high-surface-area host materials, thus being considered as a structural analog to

[1]State Key Laboratory of Chemical Engineering, Department of Chemical Engineering, Tsinghua University, 100084 Beijing, China. [2]Department of Chemistry, Tsinghua University, 100084 Beijing, China. [3]Henan Key Laboratory of Polyoxometalate Chemistry, College of Chemistry and Molecular Sciences, Henan University, Kaifeng, Henan, China. [4]CAS Engineering Laboratory for Nanozyme, Key Laboratory of Biomacromolecules, Institute of Biophysics, Chinese Academy of Sciences, 100101 Beijing, China. [5]Department of Pharmacology, School of Medicine, Institute of Translational Medicine, Yangzhou University, 225001 Yangzhou, China. [6]College of Veterinary Medicine, Yangzhou University, 225001 Yangzhou, China. [7]Research Unit of Key Technologies for Prevention and Control of Virus Zoonoses, Chinese Academy of Medical Sciences, Changchun Veterinary Research Institute, Chinese Academy of Agricultural Sciences, 130000 Changchun, China. [8]These authors contributed equally: Beibei Li, Ruonan Ma, Lei Chen, Caiyu Zhou. ✉e-mail: gaolizeng@ibp.ac.cn; niuzq@tsinghua.edu.cn

metalloenzyme. The development of Fe−N/C SACs to imitate the iron heme group of horseradish peroxidase (RHP) is a typical example. Several single-atom nanozymes with $FeN_4$ moiety have shown considerable peroxidase-like (POD-like) activity[24,25]. Tuning the metal coordination environment from $FeN_4$ to $FeN_3P$ resulted in an impressive POD-like activity comparable to natural RHP[26]. Replacing iron in $FeN_x$ moiety by other metals (e.g., Co, Mn, Zn, Mo, Ce, Pd etc.) expanded the landscape of enzyme-like properties to superoxide dismutase (SOD), catalase (CAT), glutathione peroxidase (GPx), phosphatase (PPA), and oxidase (OXD) activities[27–32]. Despite of these progresses, the range of biochemical reactions catalyzed by single-atom nanozymes is still limited. It is worth noting that many metalloenzymes contain polymetallic sites to facilitate the adsorption and activation of substrates, such as catechol oxidase ($Cu_2$), hemerythrin ($Fe_2$), cytochrome c oxidase ($Fe/Cu_2$), and so on. In light of this, introducing a second metal into SAC to form paired metal sites may realize reactions previously inaccessible to single-atom nanozymes.

In this work, we demonstrate that $Fe_2$ diatomic catalyst (DAC) exhibits lipoxidase-like (LOX-like) activity that can be used for efficient inactivation of enveloped virus. The $Fe_2$ DAC is constructed using an encapsulation-pyrolysis approach, wherein a binuclear iron complex is employed as the precursor to mediate the formation of $Fe_2$ moiety. We show that the as-obtained $Fe_2$ DAC possesses unusual LOX-like activity and can effectively destruct the influenza viral envelope via lipid peroxidation, thereby inactivating influenza viruses of different subtypes, such as H1N1 and H9N2. The advantage of nanozyme over natural enzyme is demonstrated by the limited antiviral performance of the natural LOX. Theoretical calculations indicate that the LOX-like activity originates from the low activation barrier of *cis,cis*-1,4-pentadiene moiety in vesicle membrane on binuclear iron sites. As a proof-of-concept demonstration, we incorporate the $Fe_2$ DAC onto the replacement filter of air purifier and show that the airborne influenza virus is efficiently inactivated in situ.

## Results

### Synthesis and characterization of $Fe_2$ DAC

$Fe_2$ DAC was synthesized via a macrocyclic precursor-mediated encapsulation-pyrolysis approach (Fig. 1a)[33]. Firstly, a Roberson-type binuclear iron complex ($Fe_2L$) was encapsulated inside zeolitic imidazolate framework-8 (ZIF-8) to form a composite ($Fe_2L@ZIF-8$). The encapsulation was verified by a filtrate test (see "Methods" and Supplementary Fig. 1a). The guest $Fe_2L$ molecule (<1.5 nm) is larger than the window size of host ZIF-8 (ca. 0.34 nm), thereby preventing the aggregation of the complex in adjacent cages of ZIF-8. The powder X-ray diffraction (PXRD) patterns of $Fe_2L@ZIF-8$ and pristine ZIF-8 are almost identical (Supplementary Fig. 1b), suggesting the incorporation

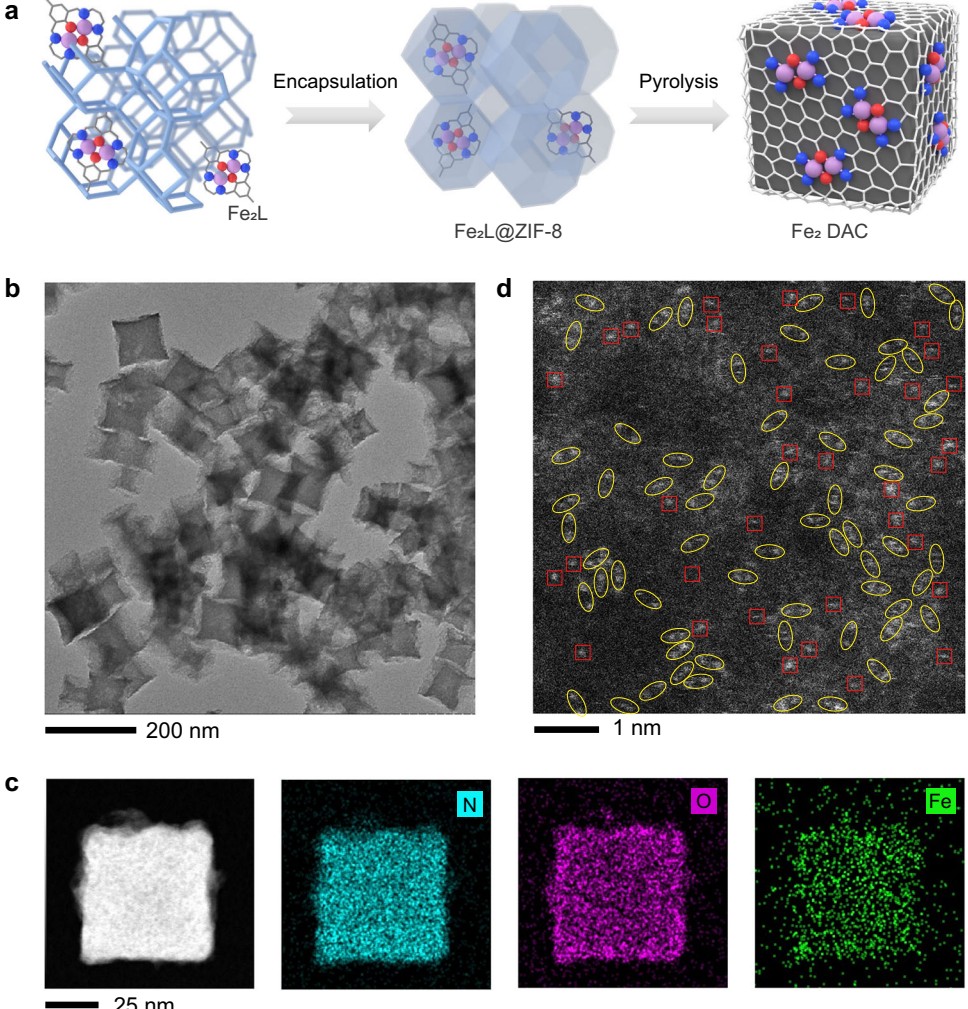

**Fig. 1 | The synthesis of Fe₂ DAC nanozyme. a** Schematic illustration of the macrocyclic-precursor Fe₂L mediated synthesis of Fe₂ DAC. **b** TEM image of Fe₂ DAC. **c** Energy-dispersive X-ray elemental mapping of Fe₂ DAC. **d** AC HAADF-STEM images of Fe₂ DAC, wherein the yellow ellipses mark the metal pairs. Three times each experiment was repeated independently with similar results, and representative images are presented.

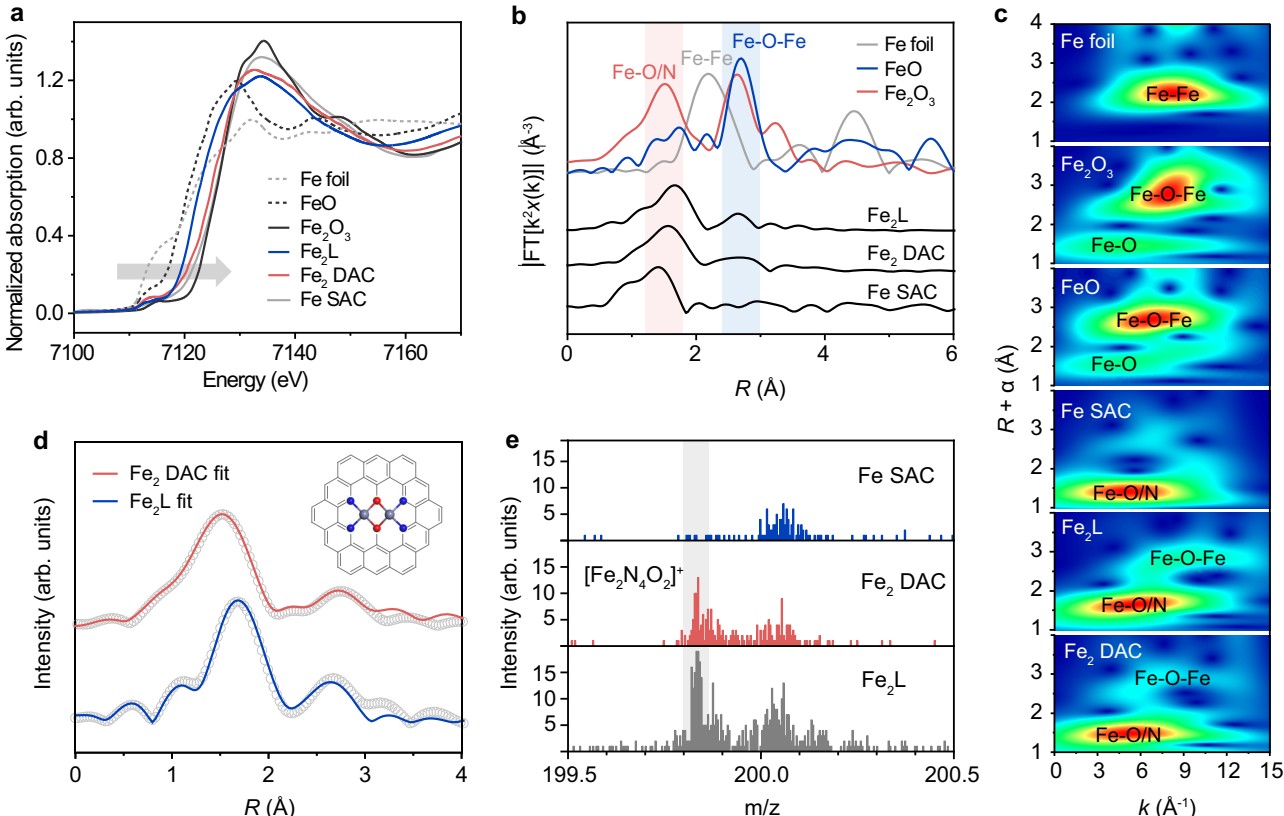

**Fig. 2 | The characterization of Fe₂ DAC nanozyme. a–c** XANES (**a**), FT EXAFS (**b**), and WT EXAFS (**c**) spectra of Fe₂ DAC, Fe SAC, and other references. **d** The FT EXAFS fitting of Fe₂ DAC and Fe₂L at the *R*-space. Inset shows the Fe₂N₄O₂ model that used for data fitting for Fe₂ DAC (C: gray; N: blue; O: red; Fe: pale purple). **e** The TOF-SIMS spectra of Fe SAC, Fe₂ DAC, and Fe₂L.

of Fe₂L has negligible effect on the purity and crystallinity of ZIF-8. Then, the Fe₂L@ZIF-8 composite was pyrolyzed at 900 °C for two hours under flowing nitrogen to yield Fe₂ DAC. Previous mechanistic study has demonstrated that the encapsulated macrocyclic precursor can preserve the Fe−O−Fe motif to a large extent during the pyrolysis process[33–35]. Meanwhile, the vast majority of the zinc in ZIF-8 was evaporated during this high-temperature treatment, which promoted the generation of porous structures in ZIF-derived carbon[36]. For comparison, Fe SAC was also prepared following the same method except that the binuclear iron precursor was replaced by iron nitrate (Supplementary Fig. 2a, see "Methods" for more details).

The as-obtained Fe₂ DAC was thoroughly characterized by different methods. Scanning electron microscopy (SEM) and transmission electron microscopy (TEM) images reveal that the cubic morphology of ZIF-8 was well-maintained throughout the encapsulation-pyrolysis process (Supplementary Fig. 2b, c and Fig. 1b). The diameters of the as-prepared Fe₂ DAC are in the range of 50–70 nm as determined by dynamic light scattering (DLS) (Supplementary Fig. 1c). Energy dispersive spectroscopy (EDS) mapping shows a uniform distribution of C, N, O, and Fe elements through the entire Fe₂ DAC particle (Fig. 1c). The Fe contents of Fe₂ DAC and Fe SAC were quantified by inductively coupled plasma optical emission spectrometry (ICP-OES), which are 1.19 wt% and 1.26 wt%, respectively. The PXRD patterns of Fe₂ DAC and Fe SAC exhibit two broad diffractions around 23° and 44°, which correspond to graphitic carbon (Supplementary Fig. 1d). The absence of characteristic diffractions of metallic iron and iron oxide implies that the iron species in Fe₂ DAC are in atomic dispersion. Aberration-corrected high-angle annular dark-field scanning transmission electron microscopy (AC HAADF-STEM) reveals that metal species present in the form of isolated bright dots (Fig. 1d), confirming their atomic dispersion on the carbon matrix. Many of the

bright dots can be marked as metal pairs (highlighted by yellow ellipses in Fig. 1d). The X-ray photoelectron spectroscopy (XPS) was further performed. The XPS spectra of Fe 2p₃/₂ peak envelop for Fe₂ DAC can be deconvoluted into three peaks (Supplementary Fig. 3). The peaks at 707.72, 709.45 and 712.78 eV can be assigned to zero-valent iron, ferrous iron (Fe²⁺), and ferric iron (Fe³⁺), respectively[37–39]. The Fe²⁺ and Fe³⁺ species are observed for both Fe₂ DAC and Fe SAC. However, the Fe²⁺/Fe³⁺ ratio in Fe₂ DAC (0.65) is much higher than that of Fe SAC (0.21), evidencing ferrous iron is the dominant species in Fe₂ DAC.

The chemical state and coordination environment of the iron sites in Fe₂ DAC were further investigated by X-ray absorption spectroscopy (XAS). Fig. 2a shows the X-ray absorption near-edge structure (XANES) spectra of the Fe *K*-edge for Fe₂ DAC and references. The absorption edge of iron in Fe₂ DAC, Fe SAC, and Fe₂L are located between those of FeO and Fe₂O₃, indicating that the oxidation state of Fe in these samples are between +2 and +3. Compared with Fe SAC, the absorption edge of Fe₂ DAC and Fe₂L are closer to that of FeO, suggesting the iron sites in Fe₂ DAC and Fe₂L have relatively lower oxidation state. These observations are in line with XPS results. The coordination environment was then studied by the Fe *K*-edge Fourier transformed extended X-ray absorption fine structure (EXAFS). As shown in Fig. 2b, the Fe *K*-edge EXAFS of Fe₂ DAC displays a strong peak located at 1.56 Å, which can be attributed to the backscattering between Fe and N/O atoms. The Fe−N/O scattering for Fe₂L shifts to higher *R*-space by 0.10 Å. This can be explained by the axial chlorine ligand in the Fe₂L complex (Supplementary Table 1)[33]. A distinct peak at 2.68 Å was observed for Fe₂ DAC and Fe₂L (Fig. 2b). This peak agrees well with the Fe−O−Fe scattering path of Fe₂O₃ (2.65 Å) and is far away from the Fe−Fe scattering path of Fe foil (2.18 Å). Notably, the Fe−O−Fe scattering path was not observed for Fe SAC. Given the similar Fe contents in Fe₂ DAC (1.19 wt%) and Fe SAC (1.26 wt%), the presence of Fe−O−Fe scattering

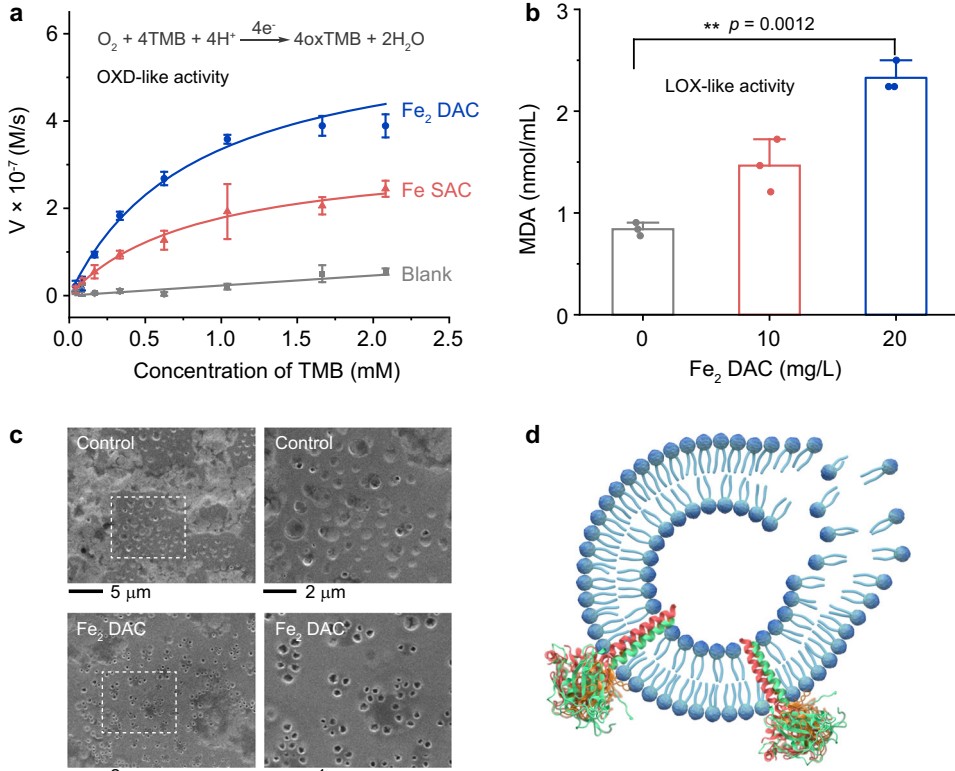

**Fig. 3 | Enzyme-like activity of Fe₂ DAC. a** Kinetics for OXD-like activity of Fe₂ DAC, Fe SAC and blank (decarbonized ZIF-8), respectively. Data are presented as means ± SD ($n = 3$ independent measurements). **b** The level of lipid peroxidation (MDA detection) after liposomes were treated by Fe₂ DAC. Data are presented as means ± SD ($n = 3$ independent measurements). The significant difference was evaluated by a two-tailed unpaired t-test. **\*\*$p < 0.01$. c** SEM images of liposomes treated by Fe₂ DAC (500 μg/mL). Control was performed with no catalyst added. **d** Schematic illustration of lipid destruction upon treatment with Fe₂ DAC.

path in Fe₂ DAC provides strong evidence to Fe−O−Fe motif in Fe₂ DAC. In addition to the EXAFS spectra in $R$-space, the Fe−O/N and Fe−O−Fe scattering paths are also visually reflected from the wavelet transform (WT) analysis of the EXAFS data for Fe₂ DAC (Fig. 2c).

We then performed quantitative EXAFS fitting to investigate the coordination configuration. The best fit shows that Fe₂ DAC adopts a Fe₂L-like Fe₂N₄O₂ configuration. As shown in Fig. 2d, the simulated EXAFS spectrum of Fe₂ DAC matches well with the experimental data based on the Fe₂N₄O₂ model (inset in Fig. 2d). According to the fitting results (Supplementary Table 2), the Fe−N and Fe−O at the first coordination shell have an interatomic distance of 2.06 Å and a total coordination number of 4.3, while Fe−Fe at the second coordination shell has an interatomic distance of 3.14 Å and a coordination number of 1.2.

In addition to EXAFS results, the coordination configuration in Fe₂ DAC is also evidenced by time-of-light secondary ion mass spectrometry (TOF-SIMS) measurement. Figure 2e displays the molecular fragments knocked out from the Fe₂ DAC, Fe SAC, and Fe₂L complex. A prominent mass fragment at m/z 199.84 which corresponds to [Fe₂N₄O₂]⁺ was observed for both Fe₂L and Fe₂ DAC. In comparison, this characteristic peak was not detected for Fe SAC. These observations together with EXAFS and HAADF-STEM provide convergent evidence to the diatomic iron sites in Fe₂ DAC.

**LOX-like activity and destruction of viral envelope**
We next investigated the intrinsic enzymatic activities of Fe₂ DAC. After extensive screening, we uncovered that Fe₂ DAC can functionally mimic a variety of enzymes, including POD, SOD, CAT, OXD, and LOX under different conditions. Generally, Fe₂ DAC exhibited enhanced catalytic activity compared with Fe SAC toward all investigated reactions. In specific, the POD-like activity was determined from 3,3′,5,5′-

tetramethylbenzidine (TMB) oxidation with H₂O₂ as the oxidant under acidic condition (pH 3–6), as shown in Supplementary Fig. 4a, b and Supplementary Table 3. The catalytic efficiency of Fe₂ DAC ($k_{cat}/K_m = 6.81 \times 10^6$ M⁻¹ s⁻¹) was about 6.5 times higher than that of Fe SAC ($k_{cat}/K_m = 1.05 \times 10^6$ M⁻¹ s⁻¹). We then calculated the POD-like specific activities for Fe₂ DAC, Fe₂L complex, and Fe SAC (Supplementary Fig. 4c), which were 73.24, 5.83, and 10.80 U mg⁻¹, respectively. In addition, Fe₂ DAC also exhibited significantly enhanced SOD- and CAT-like activities relative to Fe SAC under neutral pH 7–8 as displayed in Supplementary Fig. 5.

We subsequently investigated the OXD-like activity of Fe₂ DAC. The OXD-like activity was studied using TMB as a model substrate. Although both inducing TMB oxidation at pH 4.5, the catalytic efficiency of Fe₂ DAC ($k_{cat}/K_m = 3.47 \times 10^2$ M⁻¹ s⁻¹) was two times higher than that of Fe SAC ($k_{cat}/K_m = 1.71 \times 10^2$ M⁻¹ s⁻¹), indicating that Fe₂ DAC is more able to interact with O₂ (Fig. 3a and Supplementary Table 4). Such OXD-like catalysis under acidic pH is very similar to those discovered in other reported nanozymes[28,40,41]. In addition to TMB oxidation, we unexpectedly found that Fe₂ DAC induced lipid peroxidation when lipid substrate was present under neutral pH, which was termed as LOX-like activity. The LOX-like activity of Fe₂ DAC was investigated by incubating the material with liposome that composed of egg phospholipid, and quantified the amount of maleic dialdehyde (MDA) produced from lipid peroxidation. As shown in Fig. 3b, liposome incubated with Fe₂ DAC (20 μg mL⁻¹) for two hours resulted in increased MDA levels up to 2.7 folds compared with untreated liposome, suggesting that Fe₂ DAC can catalyze lipid peroxidation (Supplementary Fig. 6). The MDA level observed in the blank control could be due to lipid oxidation upon exposure to air. The morphology evolution of liposome was further followed by SEM. Each round pit in the SEM images represents a liposome (Fig. 3c). The structure of liposome

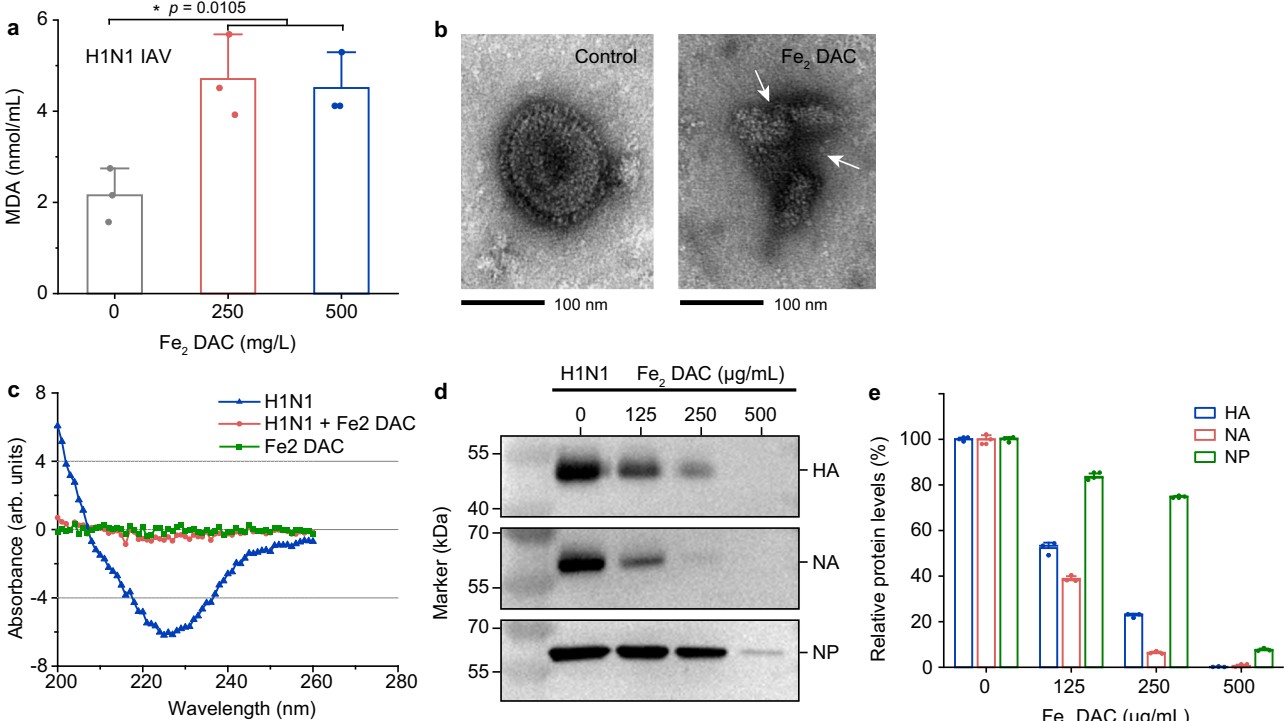

**Fig. 4 | Fe$_2$ DAC compromises the viral envelope and neighboring proteins of IAVs through lipid peroxidation. a** The level of lipid peroxidation (MDA detection) when H1N1 IAVs were treated by Fe$_2$ DAC. Data are presented as means ± SD ($n$ = 3 independent measurements). The significant difference was evaluated by a two-tailed unpaired $t$ test. *$p$ < 0.05. **b** TEM image of IAVs treated by Fe$_2$ DAC (500 μg/mL) for 90 min at RT. **c** Circular dichroism analysis of the protein structure of IAVs treated by Fe$_2$ DAC for 90 min at RT. **d** Western blot analysis of hemagglutinin (HA), neuraminidase (NA), and nucleoprotein (NP) proteins of IAVs treated by Fe$_2$ DAC for 90 min at RT. **e** Quantification of the HA, NA and NP protein levels of H1N1 IAVs after treatment with Fe$_2$ DAC. Data are presented as means ± SD ($n$ = 3 independent measurements).

was severely disrupted after incubation with Fe$_2$ DAC (Fig. 3c, d), whereas Fe SAC has little effect on the structure of the liposome (Supplementary Fig. 7). Supplementary Fig. 8 shows the quantitative statistics of liposome after treated by Fe$_2$ DAC, Fe SAC and LOX. The broken liposome accounted for 69% after Fe$_2$ DAC treatment, higher than that of LOX treatment (57%). Fe SAC treatment resulted in 34% liposome broken, which was close to the blank control (29%). This stark contrast implies that the presence of neighboring iron sites endows distinctive reactivity for the lipid peroxidation.

Given that viral envelopes are made of lipid bilayers, the LOX-like activity of Fe$_2$ DAC can be used to destruct the envelope of H1N1 IAV. We examined the MDA levels after incubation Fe$_2$ DAC with H1N1 IAV. The Fe$_2$ DAC at 250 μg mL$^{-1}$ with a two-hour incubation led to increased MDA levels up to two folds relative to untreated H1N1 IAV (Fig. 4a). The structure of IAV after Fe$_2$ DAC treatment was investigated by TEM. The viral envelope of Fe$_2$ DAC-treated IAV was considerably damaged (Fig. 4b and Supplementary Fig. 9). Circular dichroism spectra show that the absorption intensity of Fe$_2$ DAC-treated IAV is almost indistinguishable from the background (Fig. 4c), evidencing that the envelope of Fe$_2$ DAC-treated IAV was destructed. We further examined the neighboring proteins of IAV, including hemagglutinin (HA), neuraminidase (NA) and nucleoprotein (NP). As shown in Fig. 4d, e, HA, NA, and NP proteins were disrupted in a dose-dependent manner by Fe$_2$ DAC. Treatment with 500 μg mL$^{-1}$ Fe$_2$ DAC resulted in undetectable levels of HA and NA proteins, indicating their complete degradation, which is reasonable due to their close association with the lipid envelope. While the destructive effect of Fe$_2$ DAC on NP protein is weaker than that of HA and NA, which may be because NP protein is spatially away from the envelope. To exclude the interference of Fe$_2$ DAC to the western blot (WB) technique, relevant controlled trials were also performed (Supplementary Fig. 10). Previous studies have

suggested that the disruption of the viral proteins could be attributed to the attack of free radicals generated by lipid peroxidation[19,42]. Here, a similar mechanism was confirmed by electron paramagnetic resonance (EPR) spectrometry, which revealed that peroxyl radical, superoxide radical, and hydroxyl radical could be detected only in the presence of Fe$_2$ DAC and liposome (Supplementary Fig. 11). Taken together, these results indicate that Fe$_2$ DAC can cause lipid peroxidation to destruct the viral envelope.

## Non-specific antiviral effect and application in air filter
Considering that the envelope destruction can lead to virus inactivation, we then examined the virucidal efficacy of Fe$_2$ DAC. HA titer and TCID$_{50}$ assay were performed to evaluate the activity of HA and the infectivity of H1N1 virus after treatment with Fe$_2$ DAC. The results show that the HA titer of the purified H1N1 virus treated with Fe$_2$ DAC (62.5 μg mL$^{-1}$) for 15 min decreased by 2 (Fig. 5a). The HA titer and TCID$_{50}$ values of the purified H1N1 virus treated with Fe$_2$ DAC (125 μg/mL) for 90 min were reduced to 0 (Fig. 5a and Supplementary Fig. 12a). In addition, after incubation of Fe$_2$ DAC (125 μg/mL) with cell-derived H1N1 virus for 90 min, the HA titer of the virus decreased by 3 and the TCID$_{50}$ value also decreased significantly (Supplementary Fig. 12b, c).

Compared with Fe SAC and natural lipoxidase (LOX), Fe$_2$ DAC showed superior antiviral efficiency. As shown in Fig. 5a, Fe$_2$ DAC at 62.5 μg/mL decreased HA titer from 5 to 0 after 90 min treatment of H1N1, while LOX reduced less than 1 of HA titer under the same condition (Supplementary Fig. 12d). The low antiviral performance of natural LOX may be caused by different reasons. One plausible explanation is the inactivation of natural LOX caused by the attack of free radicals generated during lipid peroxidation (Supplementary Fig. 11). Another possibility is the poor interaction between LOX and the viral envelop. LOX catalyzes the peroxidation of not only free fatty

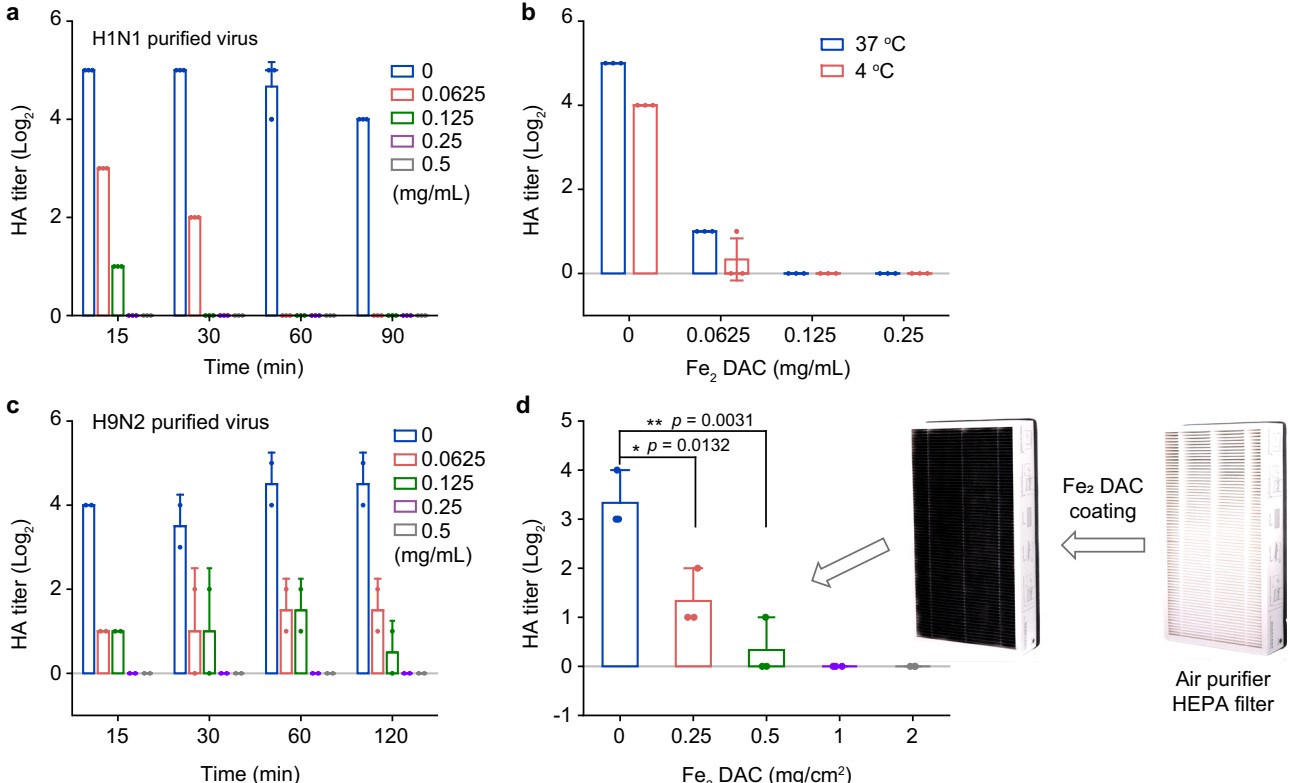

**Fig. 5 | The nonspecific antiviral effect of Fe₂ DAC and application in air filter for the inactivation of airborne flu virus. a** HA titer of Fe₂ DAC-treated H1N1 (purified virus) IAVs, under 15/30/60/90 min. **b** HA titer of Fe₂ DAC-treated H1N1 (purified virus) IAVs at 37 °C and 4 °C, under 90 min. **c** HA titer of Fe₂ DAC-treated H9N2 (purified virus) IAVs, under 15/30/60/120 min. **d** HA titer of H1N1 IAVs treated with Fe₂ DAC coated nonwoven HEPA filter, under 90 min. All the data are presented as means ± SD ($n = 3$ independent measurements). The significant difference was evaluated by a two-tailed unpaired $t$ test. *$p < 0.05$ and **$p < 0.01$.

acids but also other complex substrates like intact bio-membranes[43,44]. But the oxygenation rate of the latter is much lower than that of the free substrates[44]. Similar to bio-membranes, viral envelope consists of highly packed lipid bilayer which has limited degrees of freedom, leading to the unsatisfactory antiviral performance. To evaluate the antiviral activity of Fe SAC, we also performed HA assay (Supplementary Fig. 12e, f). The results show that Fe SAC reduced the HA titer of purified H1N1 virus rather than cell-derived virus. The HA titer of purified H1N1 virus was reduced from 5 to 2 when IAVs were treated with 500 μg/mL of Fe SAC for 90 min (Supplementary Fig. 12e). However, the HA titer of cell-derived virus had little change under the same Fe SAC treatment (Supplementary Fig. 12f).

To observe the proliferation of Fe₂ DAC-treated virus in M90 cells, we examined the intracellular NP protein. To measure NP protein, virus samples were incubated with M90 cells for one hour and then labeled with fluorescent anti-NP antibody for flow cytometer assay. As shown in Supplementary Fig. 13, the value of mean fluorescence intensity (MFI) of NP protein showed a significant reduction. This result indicates that Fe₂ DAC can reduce the infectivity of the virus. To evaluate the stability of Fe₂ DAC, the viability of the virus after Fe₂ DAC treatment was examined at 37 °C and 4 °C, respectively. The results show that the HA titer of the H1N1 virus dropped to 0 after incubation of Fe₂ DAC (62.5 μg/mL) with H1N1 virus at 37 °C or 4 °C for 90 min (Fig. 5b). This indicates that antiviral activity of Fe₂ DAC can work in a wide temperature range.

Since Fe₂ DAC inactivates H1N1 virus by destructing its envelope, we reasoned that it could inactivate other enveloped viruses through the same mechanism. To demonstrate this, we treated the purified H9N2 virus with Fe₂ DAC and performed HA titer assay. The results showed that the HA titer of the purified H9N2 virus treated with Fe₂ DAC (62.5 μg/mL) for 15 min decreased by 3 (Fig. 5c). The HA titer value

of the purified H9N2 virus treated with Fe₂ DAC (250 μg/mL) for 15 min were reduced to 0 (Fig. 5c). Besides, Fe₂ DAC also showed good anti-viral effect against other enveloped viruses such as Newcastle disease virus (NDV), SARS-CoV-2, and vesicular stomatitis virus (VSV) (Supplementary Fig. 14). However, Fe₂ DAC did not the affect non-enveloped viruses such as rotavirus and porcine circovirus (PCV) at all (Supplementary Fig. 15). The results indicate that Fe₂ DAC has broad-spectrum antiviral property and can be used for the inactivation of other enveloped viruses.

The high efficiency and wide operating temperature range of Fe₂ DAC make it more competent for real-world applications than natural LOX, which has low antiviral performance and is easily denatured due to complex environmental factors. Commercial air purifiers can only capture airborne viruses, thus having risks of secondary pollution by viral aerosol[45]. As a proof-of-concept, we coated different amounts of Fe₂ DAC onto the replacement filter of air purifier for in-situ viral disinfection. As shown in Fig. 5d and Supplementary Fig. 16a, the HA and TCID₅₀ titers were 3.33 ± 0.47 and 3.57 ± 0.01 respectively when having H1N1 virus sprayed on the air filter for 90 min. In comparison, the incorporation of Fe₂ DAC onto the filter significantly reduced both the HA and TCID₅₀ titers of H1N1 virus. In specific, with IAVs sprayed onto the outermost layer, Fe₂ DAC at a concentration of 1 mg cm⁻² reduced both the HA and TCID₅₀ titers of the virus to zero after a two-hour incubation under ambient conditions. In addition, the antiviral effect of Fe₂ DAC was not affected whether it was coated on nonwoven or gauze (Supplementary Fig. 16).

## Mechanistic insights into the LOX-like activity
Density functional theory (DFT) calculations including the solvent interactions were performed to provide mechanistic insights into the LOX-like activity of Fe₂ DAC. Based on the experimental results, a

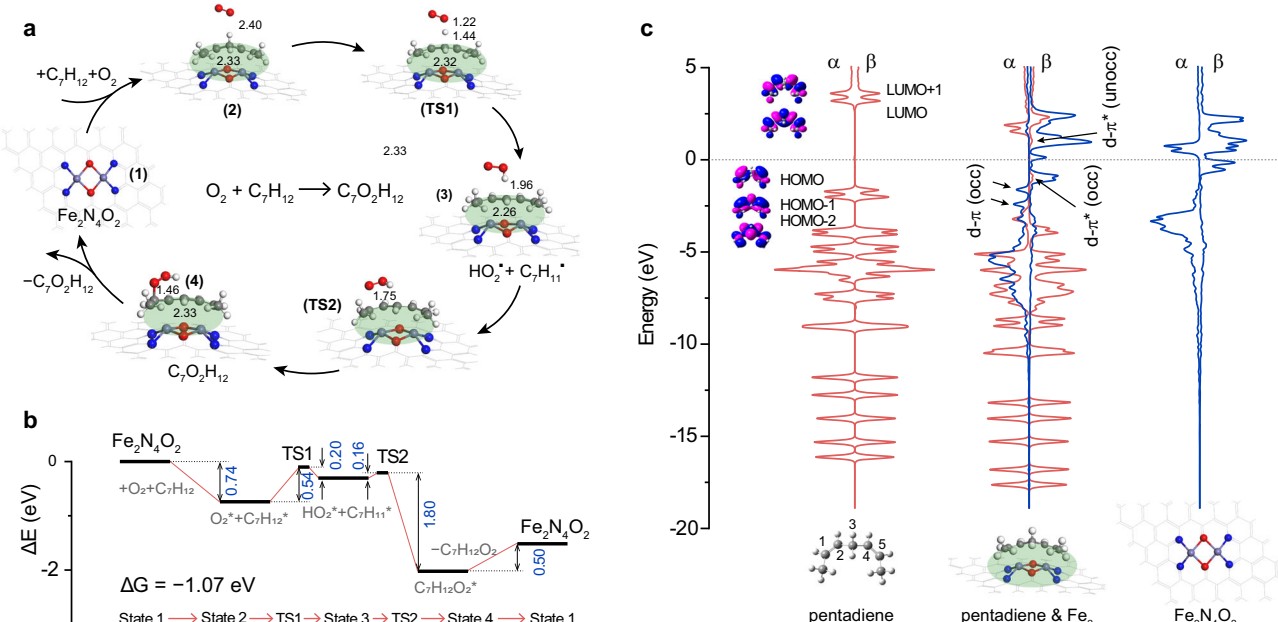

**Fig. 6 | Molecular mechanism of LOX-like enzymatic activity. a** The proposed reaction pathway of LOX-like activity, involving the dioxygenation of a substrate containing a *cis,cis*-1,4-pentadiene moiety on $Fe_2N_4O_2$ model. Bond distance unit: Å. **b** Gibbs free energy profile for key intermediate and transition states in the LOX-like catalytic cycle. Energy unit: eV. **c** Electronic structure analysis of projected electronic densities of states (pDOS) of a *cis,cis*-1,4-pentadiene moiety, $Fe_2$ DAC structure, and their interaction configuration. The asterisk (*) is used to mark species adsorbed on $Fe_2$ DAC.

$Fe_2N_4O_2$ model was selected as the active sites and a *cis,cis*-1,4-pentadiene moiety was employed as the substrate of the LOX-like catalytic activity. Based on the catalytic cycle corresponding to the dioxygenation of a substrate containing a *cis,cis*-1,4-pentadiene moiety by natural LOX enzymes[46], a four-step mechanism was constructed to describe the LOX-like activity of $Fe_2$ DAC nanozyme as shown in Fig. 6a. And the four-step mechanism can be described by the following Eqs. 2–5 for the LOX-like activity of $Fe_2$ DAC:

$$O_2 + C_7H_{10} \rightarrow C_7H_{10}O_2 \tag{1}$$

$$C_7H_{10} + Fe_2 DAC \rightarrow C_7H_{10}{}^* \tag{2}$$

$$O_2 + C_7H_{10}{}^* \rightarrow HO_2{}^{\bullet *} + C_7H_9{}^{\bullet *} \tag{3}$$

$$HO_2{}^{\bullet *} + C_7H_9{}^{\bullet *} \rightarrow C_7H_{10}O_2{}^* \tag{4}$$

$$C_7H_{10}O_2{}^* \rightarrow C_7H_{10}O_2 + Fe_2 DAC \tag{5}$$

As depicted in Fig. 6a, DFT calculations suggest that the whole catalytic cycle contains four stable states (state 1, 2, 3, 4) and two transition states (TS1, TS2). The corresponding Gibbs free energy profile is shown in Fig. 6b. The dominant elementary reactions presented in Eqs. 2–5 correspond to four steps in Fig. 6a: (1) substrate adsorption described by Eq. 2 corresponding to state 1 to state 2; (2) hydrogen abstraction and radical re-arrangement described by Eq. 3 corresponding to state 2 to state 3; (3) peroxyl radical insertion described by Eq. 4 corresponding to state 3 to state 4; and (4) product desorption described by Eq. 5 corresponding to state 4 to state 1. The adsorption of the *cis,cis*-1,4-pentadiene moiety on $Fe_2N_4O_2$ leads to its activation. The hydrogen abstraction at C-3 yields a carbon-centered bis-allylic radical $C_7H_{11}{}^{\bullet *}$, and the hydrogen atom is simultaneously transferred to the dioxygen forming peroxyl radical $HO_2{}^{\bullet *}$. The first transition state (TS1) has the highest energy barrier of 0.54 eV. After

TS1, the carbon-centered radical is rearranged forming a conjugated diene and the C-3 atom transfers from C-$sp^3$ to C-$sp^2$. The resulting peroxyl radical $HO_2{}^{\bullet *}$ migrates over the neighboring double bond and the re-arranged carbon radical is attacked by peroxyl radical $HO_2{}^{\bullet *}$, forming a 1-peroxyl-3-5-diene pentadiene moiety. Throughout the proposed reaction pathway, the change of corresponding Gibbs free energy is −1.07 eV, which indicates the catalytic cycle is feasible and easy to occur. To elucidate the bonding nature of the species involved in the mechanism, we calculated the densities of states (DOS) of the *cis,cis*-1,4-pentadiene moiety, $Fe_2N_4O_2$, and their interaction configuration (Fig. 6c). Electronic structure analysis indicates that the energy levels of $Fe_2$ minority β-spin d orbitals and pentadiene moiety π* orbitals are well matched, leading to partial occupation of the formed d-π* orbitals below Fermi level and partial unoccupation of the formed d-π* orbitals above Fermi level (electron from $Fe_2$ β-spin occupied d orbitals to pentadiene moiety unoccupied π* orbitals). In addition, the energy levels of $Fe_2$ minority α-spin d orbitals and pentadiene moiety π orbitals are well matched, leading to partial occupation of the formed d-π orbitals below Fermi level (electron from pentadiene moiety occupied π orbitals to $Fe_2$ α-spin unoccupied d orbitals). These two aspects can effectively weaken the bonding effect inside the pentadiene moiety and are responsible for the activation of substrate.

## Discussion
In summary, we prepared a $Fe_2$ DAC nanozyme through a macrocyclic precursor-mediated encapsulation-pyrolysis approach. The diatomic iron motif was identified by the combined capacities of HAADF-STEM, EXAFS, and TOF-SIMS. We demonstrated the multienzymtic activities of the $Fe_2$ DAC. Particularly, the $Fe_2$ DAC has shown distinctive activity toward lipid peroxidation, which was not observed on the Fe SAC counterpart, suggesting the uniqueness of the neighboring iron sites. DFT calculations suggest that the Fe-O-Fe motif can effectively activate the *cis,cis*-1,4-pentadiene moiety in lipid by matching the energy levels of the $Fe_2$ minority β-spin d orbitals and pentadiene moiety π* orbitals. The LOX-like activity of $Fe_2$ DAC makes it possible to destruct the envelope of influenza viruses and thus efficiently reduce their HA titer and $TCID_{50}$ values over a wide temperature range. This strategy has

great potential for the inactivation of many other enveloped viruses. Compared with natural LOX, the higher antivirus efficiency of Fe$_2$ DAC may come from its more open active sites. As a proof-of-concept, we showed that the Fe$_2$ DAC can be integrated to air cleaning devices in the form of catalyst coating for the inactivation of airborne viruses, which provides a promising strategy for comprehensive biosecurity control.

## Methods

### Material preparation

**Synthesis of Fe$_2$L complex.** Fe$_2$L complex was prepared according to the procedure previously reported with minor modifications[47]. Typically, FeCl$_2$·4H$_2$O (0.56 g, 2.8 mmol, Aladdin, 99.0%) in boiling methanol (17.5 mL) was mixed with 1,3-diaminopropane (0.356 mL, 4.2 mmol, Aladdin, 98.0%) in boiling methanol (2.5 mL). Then, 2,6-diformyl-4-methylphenol (0.46 g, 2.8 mmol, Bidepharm, 94.0%) in boiling methanol (17.5 mL) was added into the mixture. The resulting solution was kept refluxing for 120 min. During this time dark purple crystal was precipitated. The crystalline solid was collected and washed with cold methanol to give the Fe$_2$L (0.350 g, yield 39% based on FeCl$_2$·4H$_2$O). The synthesis was performed under flowing nitrogen and methanol was strictly deoxygenated before use. The Fe$_2$L complex was obtained as crystals and used for subsequent experiments directly without further purification. Elemental Analysis (Fe$_2$C$_{26}$H$_{34}$N$_4$O$_4$Cl$_2$) (%): Theoretical: C, 48.1; H, 5.3; N, 8.6. Experimental: C, 47.7; H, 4.6; N, 9.1. IR (ATR, cm-1): 1624 (s), 1555 (s), 1435 (w), 1402 (m), 1362 (w), 1321 (m), 1275 (w), 1236 (m), 1192 (w), 1120 (m), 1074 (m), 1038 (m), 973 (w), 922 (w), 874 (w), 808 (m), 771 (m).

**Preparation of Fe$_2$ DAC nanozyme.** In a typical synthesis, zinc(II) nitrate hexahydrate (Zn(NO$_3$)$_2$·6H$_2$O, 0.84 g, 2.8 mmol, Aladdin, 98.0%), Fe$_2$L (0.010 g, 0.015 mmol) and cetyltrimethyl ammonium bromide (CTAB, 0.028 g, 0.077 mmol, J&K, 99.0%) were dissolved in deionized water (28 mL), denoted as solution A. The resultant solution A was subsequently added into solution B (aqueous solution of 2-methylimidazole, 2-MeIM, 12.71 g, 0.15 mol, 196 mL, Acros, 97%) under vigorous stirring. The mixture was stirred at room temperature for 24 h. Light orange precipitate formed during this process and was collected and washed with water and methanol, respectively, each for three times. The solid was dried in vacuum oven at 70 °C overnight. Then the powder (0.30 g) was thoroughly grinded and placed in a porcelain boat, followed by heat treatment in tube furnace at 900 °C for 2 h (ramping rate 5 °C/min) under flowing nitrogen. The Fe$_2$ DAC was obtained after cooling down. The synthesis of Fe SAC nanozyme followed the same procedure as Fe$_2$ DAC except that Fe$_2$L was replaced by iron nitrate nonahydrate (Fe(NO$_3$)$_3$, 0.016 g, 0.040 mmol, Macklin, 98.5%). The synthesis of blank reference followed the same procedure as Fe$_2$ DAC except that it didn't use any iron precursor.

### Filtration test of Fe$_2$L@ZIF-8

A filtration test was performed to confirm the encapsulation of Fe$_2$L inside ZIF-8 using UV-Vis spectroscopy (UH4150 Spectrophotometer (Direct Light Detector)). The pure Fe$_2$L complex displayed a characteristic absorption at 363 nm in dichloromethane solution. This characteristic absorption exhibited a blue shift to 358 nm with the addition of acetic acid (HAc, Sinopharm Chemical Reagent Co., Ltd., 99.5%). Once encapsulated inside ZIF-8, the Fe$_2$L@ZIF-8 was dispersed in dichloromethane and sonicated for 1 h. Then, the solid was removed by filtration and the filtrate displayed no absorbance between 300–400 nm, suggesting no Fe$_2$L in the filtrate. We then added a few drops of HAc in the dichloromethane dispersion of Fe$_2$L@ZIF-8 to destroy the structure of ZIF-8. The colorless filtrate turns to pale brown and an obvious absorption peak at 358 nm could be identified, suggesting that Fe$_2$L complex was released after ZIF-8 broke down.

### Enzyme-like activity assays

**Peroxidase (POD)-like activity and kinetic parameters analysis.** The peroxidase-like activity assays of Fe$_2$ DAC were carried out using TMB (in DMSO, 20 μL, 10 mg/mL, Sigma Aldrich) as the substrate in the presence of H$_2$O$_2$ (15 μL, 10 M, Sinopharm Chemical Reagent Co., Ltd.) in NaAc buffer (1 mL, 0.1 M, pH 4.5). The absorbance of the chromogenic reactions (652 nm for TMB) was recorded at certain reaction times via the Victor Nivo™ Multimode Plate Reader (PerkinElmer, USA).

The steady-state kinetic assays were carried out at 37 °C in NaAc solution (1 mL, 0.1 M, pH 4.5) with Fe$_2$ DAC (10 μL, 1 mg/mL) in the presence of H$_2$O$_2$ and TMB. The kinetic assays of Fe$_2$ DAC with TMB as the substrate were performed by adding H$_2$O$_2$ (15 μL, 10 M) into different amounts of TMB (in DMSO, 10 mg/mL, 1, 2, 4, 8, 15, 25, 40, 50 μL). The Michaelis–Menten constants were calculated according to the Michaelis-Menten saturation curve by GraphPad Prism 7 (GraphPad Software). For comparison, the peroxidase-like activity of Fe SAC was also measured.

**Superoxide dismutase (SOD)-like activity assays.** The SOD-like activity of Fe$_2$ DAC was evaluated using a SOD assay kit (Dojindo Laboratories, Japan). In detection system, the reaction between xanthine and xanthine oxidase (XOD) produces •O$_2^-$, which reduces WST −1 to colorimetrically detectable formazan absorbing at 450 nm. Therefore, by measuring the absorption at 450 nm, the inhibition rate of Fe$_2$ DAC on •O$_2^-$, that is, the SOD-like activity can be measured. Firstly, an aqueous solution of Fe$_2$ DAC at different final concentrations (0–1 mg/mL) was mixed with 200.0 μL WST-1 working solution in a 96-well plate. After that, 20 μL of xanthine oxidase solution was added to initiate the reaction. After incubating the plate at 37 °C for 20 min, the absorbance at 450 nm was measured using the Victor Nivo™ Multimode Plate Reader (PerkinElmer, USA). Since the absorbance is proportional to the amount of superoxide anion, SOD-like activity was obtained by quantifying the decrease in absorbance at 450 nm.

**Catalase (CAT)-like activity assays.** The CAT-like activity of Fe$_2$ DAC was measured by using a specific oxygen electrode on a multiparameter analyzer (JPSJ-606L, Leici China) to monitor the increase in dissolved O$_2$ concentration. The Fe$_2$ DAC mixed with H$_2$O$_2$ aqueous solution under gentle stirring with a final concentration of 2 μg/mL. The final concentration of H$_2$O$_2$ was 500 mM, and the total volume of the mixture was 5 mL. All reactions were carried out in deoxygenated water at 37 °C.

**Oxidase (OXD)-like activity assays.** The OXD-like activity of Fe$_2$ DAC was evaluated using TMB (in DMSO, 20 μL, 10 mg/mL, Sigma Aldrich) as the substrate in NaAc buffer (1 mL, 0.1 M, pH 4.5). The absorbance of the chromogenic reactions (652 nm for TMB) representing the OXD-like activity was recorded at certain reaction times via the Victor Nivo™ Multimode Plate Reader (PerkinElmer, USA).

**Determination of lipid peroxidation.** 20 μL of Fe$_2$ DAC, Fe SAC and LOX (1 mg/mL) were mixed with 500 μL of liposomes (4 mg/mL) for 2 h at room temperature (RT). The supernatant was collected by centrifugation at 12,000 rpm (13,800 × g) for 10 min. In addition, 20 μL of Fe$_2$ DAC (5 mg/mL, 2.5 mg/mL) were mixed with 180 μL of IAVs (H1N1, 1 mg/mL) for 90 min at RT. The supernatant was collected by centrifugation at 12,000 rpm (13,800 × g) for 10 min. As MDA level is a reliable marker of lipid peroxidation[48,49], the levels of lipid peroxidation in this experiment were determined using a commercial MDA detection kit (Nanjing Jiancheng Bioengineering Institute, Nanjing, China) following the manufacturer's protocol.

Besides, to determine the lipid peroxidation levels, BODIPY 581/591 C11 (Molecular Probes, Invitrogen) also was used. The probes

were added to the reaction system of liposomes treated by different materials according to the manufacturer's instructions. The lipid ROS level was measured by a multi-scan spectrum with excitation at 488 nm and emission at 525 nm.

### Viral inactivation test

**Virus preparation.** H1N1(A/Puerto Rico/8/34) subtype IAVs strain and Newcastle disease virus (NDV, envelope virus) were propagated in 10-day-old specific-pathogen-free (SPF) embryonic chicken eggs, and purified on a discontinuous sucrose density gradient as previously described[50]. Briefly, virus-containing allantoic fluid was clarified by centrifugation at 8000 rpm ($17,888 \times g$) for 20 min. The supernatant was then centrifuged at 27,000 rpm ($96,295 \times g$) for 2.5 h. The resulting pellet was resuspended in 0.1 M PBS. This suspension was then ultracentrifuged over a discontinuous 20 to 60% sucrose gradient. Band containing the virus based was collected and centrifuged at 27,000 rpm ($96,295 \times g$) for 2.5 h to remove the sucrose. Virus stocks were obtained by resuspending the pellet in 0.1 M PBS and then were stored at −80 °C until used. Besides, the H1N1 strain was also propagated in MDCK cells. Porcine circovirus 2 (PCV-2) was propagated in porcine kidney (PK-15) cells. SARS-CoV-2, VSV and rotavirus were propagated by Vero E6, BHK-21, and MA104, respectively. Viruses were challenged to cells and cultured at 37 °C for 3 days, the medium were collected and filtered with 0.45 μm filter membrane.

**Scanning transmission electron microscopy (SEM).** 100 μL of Fe₂ DAC, Fe SAC and LOX (10 mg/mL) were mixed with 500 μL of liposomes (4 mg/mL) for 60 min at RT. Samples were fixed with 2.5% glutaraldehyde at 4 °C overnight. The structures of liposomes were revealed using SEM (Hitachi-S4800). The statistics of spherical liposomes were calculated by Image J software.

**Transmission electron microscopy (TEM).** 20 μL of Fe SAC (5 mg/mL), LOX (20 mg/mL) and different concentrations of Fe₂ DAC were mixed with 180 μL of IAVs (H1N1, 1 mg/mL) for 90 min at RT. Samples (supernatant) were negatively stained with 2% uranyl acetate. The images of influenza virus were captured using TEM (FEI Tecnai Spirit 120 kV).

**Western blot.** 20 μL of Fe₂ DAC (5, 2.5, 1.25 mg/mL) were mixed with 180 μL of IAVs (H1N1, 1 mg/mL) for 90 min at RT. To exclude technical problems linked to the interference of Fe₂ DAC to the WB techniques, pure H1N1 IAVs, Fe₂ DAC and Fe₂ DAC-treated H1N1 IAVs without incubation also were performed as control. Samples were prepared in 1×loading buffer (Biorigin) and were heated at 100 °C for 15 min before loading to SDS-polyacrylamide gels. Then separated proteins were transferred to PVDF membrane (Millipore). Western analysis was conducted by blocking the membrane in 5% skimmed milk(LABLEAD) for 1 h and incubated with primary antibodies diluted in TBST (1:1000, anti-Neuraminidase (NA), Sino Biological; 1:1000, anti-Hemagglutinin (HA), Sino Biological; 1:1000, anti-nucleoprotein (NP), Sino Biological) overnight at 4 °C. Subsequently, the membrane was washed three times with TBST and followed by addition of secondary antibody (1:5000, HRP-conjugated goat anti-mouse IgG secondary antibody, Thermo Fisher) for 1 h at RT. The blots were scanned using a fully automated chemiluminescence image analysis system (Tanon-4600SF). Quantification of the results was performed using the Image J software. For full scan blots, see the Source Data file (Figshare : https://doi.org/10.6084/m9.figshare.22651555).

**Determination of protein structure.** 20 μL of Fe₂ DAC (5 mg/mL) were mixed with 180 μL of IAVs (H1N1, 1 mg/mL) for 90 min at RT. The structural change of the viral protein was detected by circular dichroism.

**HA assay and 50% tissue culture infectious doses (TCID₅₀) detection.** To assess the antiviral activity of Fe₂ DAC, Fe SAC and LOX against enveloped virus, variable concentrations of Fe₂ DAC, Fe SAC and LOX were mixed with the influenza virus at different times at RT. Fe₂ DAC, Fe SAC and LOX were separated by centrifugation, and the supernatant was collected to detect viral titers by HA assay and TCID₅₀ assay as previously described[19].

Briefly, to conduct HA assay, the supernatants were serially diluted two-fold from $2^{-1}$ to $2^{-11}$ in 96 "V"-shaped wells. Subsequently, the equal volume of 1% chicken red blood cells (cRBCs) suspension was added and mixed to all of the wells. The viral HA titers were evaluated after the plates had been incubated for 10 min at 37 °C.

To measure the TCID₅₀, for H1N1 virus, the supernatants were serially diluted 10-fold from $10^{-1}$ to $10^{-8}$, and each dilution ($10^{-1}$–$10^{-8}$) was inoculated into M90 cell monolayers under 37 °C for 1 h. Then, the monolayer was rinsed with PBS, overlaid with maintenance medium (1% FBS in DMEM) and incubated at 37 °C for 72 h. For influenza virus and Newcastle disease virus positive wells, the HA assay was performed. For SARS-CoV-2, VSV, rotavirus and porcine circovirus, titer assays were taken using Vero E6, BHK-21, MA104 and PK-15 cells, respectively. The cells were plated in 96-wells and infected with serial dilutions of virus in serum-free DMEM and incubated for 2 h, then replaced medium by DMEM containing 2% FBS and 1% penicillin-streptomycin, and cultured at 37 °C with 5% CO₂ for 3–5 days. The Log₁₀TCID₅₀ per 0.1 mL was calculated using the Reed-Muench method as described previously[51]. For detection of PCV-2, immunofluorescence assay was performed. PK-15 cells were infected with PCV-2, which was untreated or pretreated by nanozyme for 24 h. Then the cells were fixed with 4% paraformaldehyde for 10 min at RT. After washing with PBS, the cells were incubated for 10 min with PBS containing 0.4% Triton X-100, and washed by PBS three times. The cells were incubated at 37 °C for 1 h with a pig polyclonal antiserum (VMRD, Washington, USA) and subsequently reacted with FITC-conjugated rabbit anti-pig antibody (Southern Biotech, Birmingham, USA) for 1 h. The data were obtained using fluorescence microscope.

**Detection of NP proteins.** 20 μL of Fe₂ DAC (5, 1.25, 0.625 mg/mL) were mixed with 180 μL of H1N1 for 90 min at RT, the supernatant was collected by centrifugation at 8000 rpm ($6010 \times g$) for 10 min. The supernatants were inoculated into M90 cell monolayers by using 6-well plates under 37 °C for 1 h. After that, the monolayer was rinsed with PBS, overlaid with medium (1% FBS in DMEM) and incubated at 37 °C for another 36 h. To identify the ability of influenza viruses to replicate within cells, the NP protein assay was performed. The cells were rinsed three times with PBS, fixed with 4% paraformaldehyde (LABLEAD), saturated with 0.4% Triton X-100 (Solarbio), and then blocked with 5% albumin bovine V (BSA, LABLEAD) before staining with anti-NP (1:300, Sino Biological). The cells were washed with PBST three times and incubated with fluorescein Alexa Fluor 488-conjugated goat anti-mouse IgG (1:400, Invitrogen) at 37 °C. The data was acquired using a BD FACSCalibur flow cytometer and then analyzed with FlowJo_v10 software.

**Application of Fe₂ DAC in air cleaning system.** 25 μL of Fe₂ DAC (80 mg/mL, 40 mg/mL, 20 mg/mL, and 10 mg/mL) were coated on the nonwoven or gauze (two layers) of replacement filter of an air purifier. After drying by airflow, 200 μL IAVs was sprayed onto the outermost layer. After incubating for 2 h at RT. Samples were placed in 1.5 mL EP tubes and the supernatant was collected by centrifugation at 8000 rpm ($6010 \times g$) for 5 min, and the population of viable virus was then measured using HA and TCID₅₀ methods.

### Density functional theory (DFT) calculations

All calculations were performed within the framework of density functional theory (DFT) using a plane-wave basis set, as implemented in the

Vienna ab initio simulation package (VASP.6.1.0)[52]. A (5 × 5) graphene supercell containing 50 C atoms, 4 N atoms, 2 O atoms and 2 Fe atoms was taken as the Fe$_2$ DAC structural model. The Perdew-Burke-Ernzerhof functional of the generalized gradient approximation (GGA)[53,54] was used for geometry optimization and energy calculations. All calculations are spin-polarized with an energy cutoff of 500 eV and Gaussian smearing of 0.05 eV. A vacuum height of 15 Å was set in the vertical directions to avoid the interaction between periodic images. The Brillouin zones were sampled with the (3 × 3 × 1) Monkhorst-Pack[55] mesh k-point grids. The convergence criteria of the electronic energy and forces were set to $10^{-5}$ eV and 0.01 eV/Å, respectively. In order to better describe the catalytic mechanism, the dielectric constant (EB_K = 80) of aqueous solvents is considered to model the water environment. Transition states were calculated by the CI-NEB (climbing image-nudged elastic band) method[56], also implemented in the vasp package. During the search of transition states, a lower force threshold of "0.03 eV/Å" was used, and stretching frequencies were analyzed to ensure only one imaginary frequency for each of the transition states. The activation energies were calculated by the following equation:

$$E_a = E_{TS} - E_{IS}$$

where $E_{TS}$ is the transition state energy and $E_{IS}$ is the energy of the initial state.

### Reporting summary

Further information on research design is available in the Nature Portfolio Reporting Summary linked to this article.

## Data availability

The data generated in this study are provided in Supplementary Information/Source data file (Figshare https://doi.org/10.6084/m9.figshare.22651555).

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

## Acknowledgements

This work was supported by the National Key R&D Program of China (2019YFA0709200), the National Natural Science Foundation of China (22075162 and 81930050), National Natural Science Foundation of China Foundation of Innovative Research Group grant (22121003), and Tsinghua University Initiative Scientific Research Program (20221080067). We acknowledge the BL14W1 station in Shanghai Synchrotron Radiation Facility (SSRF) and 4B9A station in Beijing Synchrotron Radiation Facility (BSRF) for the collection of XAFS data.

## Author contributions

Z.N., L.G., and Y.L. conceptualized and guided this work. Z.N. and L.G. designed the experiments. B.L., Y.-X.Z., H.H., Q.H., Y.W., and Y.C. performed the material synthesis and characterization. X.W. and B.L. performed catalytic activity evaluation. R.M., C.Z., M.S., P.H., T.Q., and C.L. performed antiviral experiments. X.Z. and J.Y. performed XAS simulation. L.C. performed the DFT calculations. Z.N., L.G., B.L., R.M., and L.C. wrote the paper. All the authors participated in the data analysis and commented on the manuscript.

## Competing interests

Z.N., L.G., B.L., R.M., and Y.Z. filed a provisional patent application. The other authors declare no competing interests.
