## [Peer Review File · Nature Communications]

Reviewers' Comments:

Reviewer #1:

Remarks to the Author:

The paper by Li et al describe the virucidal activity of Iron nanozymes as potential tool for enveloped virus inactivation.

I am not qualified to judge the chemistry and material science part therefore my review will focus only on the biological assays.

The experiments show the activity on influenza virus, but if the mechanism is claimed to be aspecific, additional distinct viruses need to be added. For instance coronaviruses or paramyxoviruses belong to different families but they are relevant respiratory viruses pathogenic in humans and, if inactivated the data will support the conclusions of the authors and sustain their possible use in air filters. Moreover the addition of non-enveloped virus such as rhinovirus should be included as negative control.

In many figures, error bars are not present (for instance Figure 5, S11, S13, and others) and in the caption is not indicated how many independent experiments were performed. For reliability and significance, the experiments have to be performed in at least 3 independent experiments and the number of replicates added in each figure legend.

The TEM and circular dichroism experiments are performed with a centrifugation step in between the incubation and the visualization, this step could affect the integrity of the virions. The experiments should be performed without this intermediate step to conclude that the nanozymes disrupt the viruses.

In figure 4d is shown a lower amount of viral proteins in the Western Blot. However, the mechanism of action claimed by the authors is on lipids and not on proteins. Which is the proposed mechanism for viral proteins? To exclude technical problems linked to the interference of the nanozymes to the WB techniques the same WB performed without time of incubation should be performed.

MINOR COMMENTS:

-Page 2 line 2, list either viruses or diseases

-Clarification of the duration treatment for result 4b, 4c and 4d should be mentioned in the text or in the legend

-cell-produced H1N1 is misleading reword

-in figure S15 include a condition with only nanozymes

-reduction of virulence of virus is rather reduction of infectivity of the virus

Reviewer #2:

Remarks to the Author:

Reviewer noted high novelty of presented findings, namely, the development of iron diatomic catalyst with lipoxidase-like activity for enveloped viruses inactivation, moreover demonstrating its increased activity compared to single-atom iron and natural LOX enzyme, which is of high significance in the field of nanozymes development. The work supports the conclusion and claims, all the methods are relevant and meet expected standards in the field.

However, major revision should be made prior to publication.

The following remarks should be considered:

1) "The structure of liposome was severely disrupted after incubation with Fe₂ DAC (Fig. 3c,d), whereas Fe SAC does not affect the structure of the liposome at all (Supplementary Fig. 9)." – Fig S9 by itself doesn't allow to conclude that Fe SAC has absolutely no effect on liposome structure. It is critical to quantify these experiments prior to any conclusions as well as for other scientists to be able to interpret, compare, and reproduce it. LOX enzyme influence on liposome morphology is also necessary to include.

2) All 4 levels of proof-of-concept (MDA levels, liposome peroxidation, IAV destruction, and

virucidal activity) should be performed in a similar manner (control, Fe DAC, LOX, Fe₂ DAC) to solidify the statement of natural LOX lower antiviral activity and provide more comprehensive studies.

- 3) Mass fraction of Fe₂ DAC should be measured in relation to ZIF-derived carbon.
- 4) Fe SAC behavior on Fig. S14 should be explained in detail in the manuscript.
- 5) Authors should make more emphasis on viral proteins disruption by Fe₂ DAC including proposed mechanism and quantify these results.
- 6) Fe₂/Fe ratio was already quantified with ICP-OES, percentages calculated from AC HAADF-STEM can be misleading and redundant, especially considering authors' remarks about Zn signals rendering quantification.
- 7) Fig. S8 and Fig. 3b show notable lipid peroxidation in control experiments, which should be double-checked and explained in the manuscript.
- 8) Error bars for the results on Fig. 5a-c should be added.

To improve manuscript comprehensibility and reproducibility, further comments should be reviewed:

- 1) Characterization section in SI should be further structured.
- 2) Compound purification methods need more details for successful results reproduction.
- 3) All the synthesis steps should be visualized in more details.
- 4) All figure captions should be carefully checked for mistakes (for example, Fig. S9 claims that the presented results are for IAV), typos, and signatures without sufficient explanation.
- 5) More representative images of Fe₂ DAC-treated IAVs should be enclosed in supplementary materials in addition to Fig. 4b with a single example.
- 6) All the abbreviations should be explained when first mentioned.
- 7) "Non-specific antiviral effect and application in air filter" section should be rewritten in a clearer manner providing interpretation of any observation. LOX low antiviral activity explanation should be expanded and supplemented with relevant refs.
- 8) Figures quality should be increased.

Reviewer #3:

Remarks to the Author:

The manuscript "Diatomic Iron Nanozyme with Lipoxidase-like Activity for Efficient Inactivation of Enveloped Virus" by Li and coworkers presents an experimental and computational study of Fe₂ doped dual atom catalyst on graphitic surface as potential nano-isozyme like activity. Increasing the number of metal dopants at the active site has been shown to increase the efficiency of the catalysts and enhance the activity for the specific enzymatic mechanism. The manuscript has been well written. To characterize the structural morphology of the catalyst, the authors have performed X-ray spectroscopic studies combined with theoretical calculations. Following the DFT calculations, a mechanistic pathway has been proposed.

However, I am not convinced about the conclusions drawn from the theoretical observations. In addition, the reaction mechanism proposed in this study has a few drawbacks. Therefore, before having the final decision from my side, I would like to suggest the authors comment on the following points.

(1) The authors have characterized a four-step mechanism. The reaction proceeds with the initial TS of 0.96 eV. I believe this value would be higher for the reaction to proceed with greater yield. This is not in agreement with the experimental results, where it shows that the diatom catalyst has the better performance as the LOX substrate. Either the reaction proceeds with the alternate pathway, or some of the environmental interactions are not taken care of, such as the solvent interactions. The authors should comment if the reaction conditions are taken care of for the calculations.

(2) The authors have proposed that the diene substrate is adsorbed first and then the O₂. The authors should point out what led them to propose this mechanism. Is there any alternative mechanism possible for this process? The authors have not elucidated the different possible reaction mechanisms. It might be that O₂ molecule is activated by the diatom dopant, which reacts with the diene to give the respective product.

Reviewer #1 (Remarks to the Author):

COMMENTS

The paper by Li et al describe the virucidal activity of Iron nanozymes as potential tool for enveloped virus inactivation.

I am not qualified to judge the chemistry and material science part therefore my review will focus only on the biological assays.

Q1. The experiments show the activity on influenza virus, but if the mechanism is claimed to be aspecific, additional distinct viruses need to be added. For instance, coronaviruses or paramyxoviruses belong to different families but they are relevant respiratory viruses pathogenic in humans and, if inactivated the data will support the conclusions of the authors and sustain their possible use in air filters. Moreover, the addition of non-enveloped virus such as rhinovirus should be included as negative control.

Response:

Thank you very much for the suggestions. Results of additional distinct enveloped viruses such as SARS-CoV-2, Vesicular stomatitis virus (VSV) and Newcastle disease virus (NDV) treated with different concentration Fe₂ DAC are shown in Fig. C1, and non-enveloped viruses such as Rotavirus and porcine circovirus (PCV) are used as negative controls (Fig. C2). The results show that different enveloped viruses can be inactivated by Fe₂ DAC, but non-enveloped viruses are not affected. The results confirm the inactivation mechanism of Fe₂ DAC is non-specific. These results were added in the revised supplementary information.

Fig. C1. Antiviral effect of Fe₂ DAC to enveloped virus. Antiviral effect of Fe₂ DAC to **a**, Newcastle disease virus (NDV), **b**, SARS-CoV-2 and **c**, Vesicular stomatitis virus (VSV). **d**, Fluorescence images of BHK-21 cells infected by VSV after incubation with different concentrations of Fe₂ DAC. Scale bar = 100 μm.

Fig. C2. Antiviral effect of Fe₂ DAC to non-enveloped virus. Antiviral effect of Fe₂ DAC to **a**, Rotavirus and **b**, Porcine circovirus (PCV). **c**, Fluorescence images of PK-15 cells infected by PCV after incubation with different concentrations of Fe₂ DAC. Scale bar = 50 µm.

Q2. In many figures, error bars are not present (for instance Figure 5, S11, S13, and others) and in the caption is not indicated how many independent experiments were performed. For reliability and significance, the experiments have to be performed in at least 3 independent experiments and the number of replicates added in each figure legend.

Response:

Thanks for pointing out this issue. We have rerun the related experiments and added error bars. Figure 5 is as below, and other spectra have also been shown in the Supplementary Information. The columns that appear to have no error bars are because the three replicates are completely consistent. The raw data of Figure 5 are listed in the following tables. According to the experimental principle and methods of HA assay, the experimental data (HA titer) is the maximum dilution (Log₂) of the virus to completely agglutinate chicken red blood cells and is an integer when it is read. The parallelism of HA assay results is generally excellent. Thus, in some cases, there is still no error bar even if the determination has been repeated for three times. If there is a difference, it is usually less than 1.

Fig. 5. The nonspecific antiviral effect of Fe₂ DAC and application in air filter for the inactivation of airborne flu virus. **a**, HA titer of Fe₂ DAC-treated H1N1 (purified virus) IAVs, under 15/30/60/90 min. **b**, HA titer of Fe₂ DAC-treated H1N 1 (purified virus) IAVs at 37 °C and 4 °C, under 90 min. **c**, HA titer of Fe₂ DAC-treated H9N2 (purified virus) IAVs, under 15/30/60/120 min. **d**, HA titer of H1N1 IAVs treated with Fe₂ DAC coated nonwoven HEPA filter, under 90 min. *P < 0.05 and **P < 0.01. The error bars represent standard deviations for three independent measurements.

Fig. 5a, HA titer of Fe₂ DAC-treated H1N1 (purified virus) IAVs, under 15/30/60/90 min.

Time min	0 mg/mL	0.0625 mg/mL	0.125 mg/mL	0.25 mg/mL	0.5 mg/mL
15	5 5 5	3 3 3	1 1 1	0 0 0	0 0 0
30	5 5 5	2 2 2	0 0 0	0 0 0	0 0 0
60	5 5 4	0 0 0	0 0 0	0 0 0	0 0 0
90	4 4 4	0 0 0	0 0 0	0 0 0	0 0 0

Fig. 5b, HA titer of Fe₂ DAC-treated H1N1 (purified virus) IAVs at 37 °C and 4 °C, under 90 min.

Fe ₂ DAC mg/mL	37 °C			4 °C		
0	5	5	5	4	4	4
0.0625	1	1	1	1	0	0
0.125	0	0	0	0	0	0
0.25	0	0	0	0	0	0

Fig. 5c, HA titer of Fe₂ DAC-treated H9N2 (purified virus) IAVs, under 15/30/60/120 min.

Time min	0 mg/mL			0.0625 mg/mL			0.125 mg/mL			0.25 mg/mL			0.5 mg/mL		
15	4	4	4	1	1	1	1	1	1	0	0	0	0	0	0
30	3	4	3.5	0	2	1	0	2	1	0	0	0	0	0	0
60	4	5	4.5	1	2	1.5	1	2	1.5	0	0	0	0	0	0
90	4	5	4.5	1	2	1.5	0	1	0.5	0	0	0	0	0	0

Fig. 5d, HA titer of H1N1 IAVs treated with Fe₂ DAC coated nonwoven HEPA filter, under 90 min

Fe ₂ DAC (mg/cm ²)	HA titer (Log ₂)		
0	3	3	4
0.25	2	1	1
0.5	1	0	0
1	0	0	0
2	0	0	0

Q3. The TEM and circular dichroism experiments are performed with a centrifugation step in between the incubation and the visualization, this step could affect the integrity of the virions. The experiments should be performed without this intermediate step to conclude that the nanozymes disrupt the viruses.

Response:

Thanks for the question and thoroughness. We have added the TEM and circular dichroism (CD) experiments without a centrifugation step in between the incubation and the visualization. The lipid envelope of Fe₂ DAC-treated IAV was considerably damaged (Fig. C3). CD spectra show that the absorption intensity of Fe₂ DAC-treated IAV is almost indistinguishable from the background (Fig. C4). The results prove that Fe₂ DAC can indeed disrupt the viruses.

Fig. C3. TEM images of H1N1 IAVs treated with variable concentrations of Fe₂ DAC.

Fig. C4. Circular dichroism analysis of the protein structure of IAVs treated by Fe₂ DAC (0.5 mg/mL) for 90 min at RT without centrifugation step.

Q4. In figure 4d is shown a lower amount of viral proteins in the Western Blot. However, the mechanism of action claimed by the authors is on lipids and not on proteins. Which is the proposed mechanism for viral proteins? To exclude technical problems linked to the interference of the nanozymes to the WB techniques the same WB performed without time of incubation should be performed.

Response:

Thanks for this useful thoroughness. We have made more emphasis on the proposed mechanism of protein destruction and make corresponding additions in the revised manuscript. “Previous studies have suggested that the disruption of viral proteins could be attributed to the attack of free radicals generated by lipid peroxidation (Adv. Sci. 10, e2206869 (2023); Theranostics 9, 6920–6935 (2019)). Here, a similar mechanism was confirmed by electron paramagnetic resonance (EPR) spectrometry, which revealed that peroxy radical, superoxide radical and hydroxyl radical could be detected only in the presence of Fe₂ DAC and liposomes (Supplementary Fig.11).”

To exclude technical problems linked to the interference of Fe₂ DAC to the WB techniques, pure H1N1 IAVs, Fe₂ DAC and Fe₂ DAC-treated H1N1 IAVs without incubation were performed as control. As shown in Fig. C5, Fe₂ DAC does not affect the test of Western Blot.

Fig. C5. Western blot of HA, NA and NP proteins of H1N1 IAVs, Fe₂ DAC, and Fe₂ DAC-treated H1N1 IAVs without incubation.

MINOR COMMENTS:

Q1. Page 2 line 2, list either viruses or diseases

Response:

Thank you for pointing out the inconsistency. We have listed all the viruses.

Q2. Clarification of the duration treatment for result 4b, 4c and 4d should be mentioned in the text or in the legend

Response:

Their duration treatment is 90 min, which has been added in the revised manuscript.

Q3. cell-produced H1N1 is misleading reword

Response:

We have changed cell-produced H1N1 to cell-derived H1N1 in the revised manuscript and Supplementary Information.

Q4. in figure S15 include a condition with only nanozymes

Response:

Detection of NP protein expression has been re-measured, with Fe₂ DAC nanozymes as a control (Fig. C6).

Fig. C6. Detection of NP protein expression in M90 cells. **a**, Flow cytometry of NP protein expression in M90 cells. **b**, Mean fluorescence intensity (MFI) of NP protein expression in (a). **P < 0.01 and ***P < 0.001. The error bars represent standard deviations for three independent measurements.

Q5. reduction of virulence of virus is rather reduction of infectivity of the virus

Response:

We have changed the wording of virulence to infectivity in the revised manuscript.

Reviewer #2 (Remarks to the Author):

Reviewer noted high novelty of presented findings, namely, the development of iron diatomic catalyst with lipoxidase-like activity for enveloped viruses inactivation, moreover demonstrating its increased activity compared to single-atom iron and natural LOX enzyme, which is of high significance in the field of nanozymes development. The work supports the conclusion and claims, all the methods are relevant and meet expected standards in the field.

However, major revision should be made prior to publication.

The following remarks should be considered:

Q1. "The structure of liposome was severely disrupted after incubation with Fe₂ DAC (Fig. 3c,d), whereas Fe SAC does not affect the structure of the liposome at all (Supplementary Fig. 9)." – Fig S9 by itself doesn't allow to conclude that Fe SAC has absolutely no effect on liposome structure. It is critical to quantify these experiments prior to any conclusions as well as for other scientists to be able to interpret, compare, and reproduce it. LOX enzyme influence on liposome morphology is also necessary to include.

Response:

We thank the reviewers for their insightful comments and suggestion. We have taken a larger range of SEM pictures and performing quantitative statistics. As shown in Fig. C7, the structure of liposome was severely disrupted after incubation with Fe₂ DAC and LOX, whereas Fe SAC has little affect effect on the structure of the liposome.

Fig. C7. Destruction of liposomes. **a**, SEM images and **b**, quantitative statistics of liposomes treated by Fe₂ DAC, Fe SAC and LOX.

Q2. All 4 levels of proof-of-concept (MDA levels, liposome peroxidation, IAV destruction, and virucidal activity) should be performed in a similar manner (control, Fe DAC, LOX, Fe₂ DAC) to solidify the statement of natural LOX lower antiviral activity and provide more comprehensive studies.

Response:

We acknowledge that all the tests should be performed in a similar way for all the samples. In the revised manuscript, destruction of liposomes, the levels of MDA, and lipid ROS were tested under the same conditions. As shown in Fig. C7 and C8, Fe₂ DAC have comparable lipid peroxidation activity with LOX. The experiments on IAV destruction were also performed under appropriate conditions. Variable concentrations of Fe₂ DAC, Fe SAC and

LOX were mixed with the influenza virus at different times at RT. HA titer did not have significant change after Fe SAC and LOX treating the purified H1N1 virus (Supplementary Fig. 12d-f), suggesting their low antiviral activities. In addition, TEM images of H1N1 IAVs treated with Fe SAC (0.5 mg/mL), LOX (2 mg/mL) and variable concentrations of Fe₂ DAC also have been added (Supplementary Fig. 9), which evidences that Fe₂ DAC has higher antiviral activity than LOX.

Fig. C8. Detection of lipid peroxidation. Levels of **a**, MDA and **b**, lipid ROS (BODIPY 581/591 C11 probe) after liposomes treated by Fe₂ DAC, Fe SAC, and LOX. *P < 0.05. The error bars represent standard deviations for three independent measurements.

Q3. Mass fraction of Fe₂ DAC should be measured in relation to ZIF-derived carbon.

Response:

We have quantified the Fe contents of Fe₂ DAC and Fe SAC by inductively coupled plasma optical emission spectrometry (ICP-OES), which are 1.19 wt% and 1.26 wt%, respectively. These data are added in the revised manuscript.

Q4. Fe SAC behavior on Fig. S14 should be explained in detail in the manuscript.

Response:

Thanks for this valuable suggestion. Fig. S14 (now Fig. S12e,f) have been explained in detail in the revised manuscript. "To evaluate the antiviral activity of Fe SAC, we also performed HA assay (Supplementary Fig. 12e,f). The results show that Fe SAC reduced the HA titer of purified H1N1 virus rather than cell-derived virus. The HA titer of purified

H1N1 virus was reduced from 5 to 2 when IAVs were treated with 500 $\mu\text{g/mL}$ of Fe SAC for 90 min (Supplementary Fig. 12e). However, the HA titer of cell-derived virus had little change under the same Fe SAC treatment (Supplementary Fig. 12f).”

Q5. Authors should make more emphasis on viral proteins disruption by Fe₂ DAC including proposed mechanism and quantify these results.

Response:

We appreciate the reviewer for the suggestion. Quantitative analysis of the Western blot results of hemagglutinin (HA), neuraminidase (NA), and nucleoprotein (NP) proteins of IAVs treated by Fe₂ DAC has been added (Fig. C9) in the revised manuscript.

The protein disruption is very likely induced by free radicals generated during Fe₂ DAC treatment. As discussed in the revised manuscript, “Previous studies have suggested that the disruption of viral proteins could be attributed to the attack of free radicals generated by lipid peroxidation (Adv. Sci. 10, e2206869 (2023); Theranostics 9, 6920–6935 (2019)). Here, a similar mechanism was confirmed by electron paramagnetic resonance (EPR) spectrometry, which revealed that peroxy radical, superoxide radical and hydroxyl radical could be detected only in the presence of Fe₂ DAC and liposomes (Supplementary Fig.11).”

Fig. C9. Quantification of the HA, NA and NP protein levels of H1N1 IAVs after treatment with Fe₂ DAC.

Q6. Fe₂/Fe ratio was already quantified with ICP-OES, percentages calculated from AC HAADF-STEM can be misleading and redundant, especially considering authors' remarks about Zn signals rendering quantification.

Response:

We acknowledge that the Fe₂/Fe ratio calculated from the AC HAADF-STEM is misleading, which has been deleted from the revised manuscript.

Q7. Fig. S8 and Fig. 3b show notable lipid peroxidation in control experiments, which should be double-checked and explained in the manuscript.

Response:

We appreciate the reviewer for the question. The commercial lipid could be oxidized upon exposure to air during storage and thus already has certain level of MDA before forming liposome. The explanation has been included in the revised manuscript (Figure caption of Fig. 3b).

Q8. Error bars for the results on Fig. 5a-c should be added.

Response:

Thank you for pointing out this issue. All the related experiments have been repeated for three times independently and the error bars have been added in the revised manuscript. The columns that appear to have no error bars are because the three replicates are completely consistent. The raw data of Figure 5 are listed in the following tables.

Fig. 5. The nonspecific antiviral effect of Fe₂ DAC and application in air filter for the inactivation of airborne flu virus. **a**, HA titer of Fe₂ DAC-treated H1N1 (purified virus) IAVs, under 15/30/60/90 min. **b**, HA titer of Fe₂ DAC-treated H1N 1 (purified virus) IAVs at 37 °C and 4 °C, under 90 min. **c**, HA titer of Fe₂ DAC-treated H9N2 (purified virus) IAVs, under 15/30/60/120 min. **d**, HA titer of H1N1 IAVs treated with Fe₂ DAC coated nonwoven HEPA filter, under 90 min. *P < 0.05 and **P < 0.01. The error bars represent standard deviations for three independent measurements.

Fig. 5a, HA titer of Fe₂ DAC-treated H1N1 (purified virus) IAVs, under 15/30/60/90 min.

Time min	0 mg/mL	0.0625 mg/mL	0.125 mg/mL	0.25 mg/mL	0.5 mg/mL
15	5 5 5	3 3 3	1 1 1	0 0 0	0 0 0
30	5 5 5	2 2 2	0 0 0	0 0 0	0 0 0
60	5 5 4	0 0 0	0 0 0	0 0 0	0 0 0

90	4	4	4	0	0	0	0	0	0	0	0	0
----	---	---	---	---	---	---	---	---	---	---	---	---

Fig. 5b, HA titer of Fe₂ DAC-treated H1N1 (purified virus) IAVs at 37 °C and 4 °C, under 90 min.

Fe ₂ DAC mg/mL	37°C			4°C		
0	5	5	5	4	4	4
0.0625	1	1	1	1	0	0
0.125	0	0	0	0	0	0
0.25	0	0	0	0	0	0

Fig. 5c, HA titer of Fe₂ DAC-treated H9N2 (purified virus) IAVs, under 15/30/60/120 min.

Time min	0 mg/mL	0.0625 mg/mL	0.125 mg/mL	0.25 mg/mL	0.5 mg/mL
15	4 4 4	1 1 1	1 1 1	0 0 0	0 0 0
30	3 4 3.5	0 2 1	0 2 1	0 0 0	0 0 0
60	4 5 4.5	1 2 1.5	1 2 1.5	0 0 0	0 0 0
90	4 5 4.5	1 2 1.5	0 1 0.5	0 0 0	0 0 0

Fig. 5d, HA titer of H1N1 IAVs treated with Fe₂ DAC coated nonwoven HEPA filter, under 90 min

Fe ₂ DAC (mg/cm ²)	HA titer (Log ₂)		
0	3	3	4
0.25	2	1	1
0.5	1	0	0
1	0	0	0
2	0	0	0

To improve manuscript comprehensibility and reproducibility, further comments should be reviewed:

Q1. Characterization section in SI should be further structured.

Response:

We have amended the Supplementary information accordingly.

Q2. Compound purification methods need more details for successful results reproduction.

Response:

Thank you for the suggestion. The Fe₂L compound was obtained as crystals and used for subsequent experiments without further purification. The powder X-ray diffraction (PXRD) demonstrates the purity of the compound (Fig. C10).

Fig. C10. Powder X-ray diffraction (PXRD) patterns of Fe₂L.

Q3. All the synthesis steps should be visualized in more details.

Response:

Thanks for the helpful suggestion. We have added a schematic diagram to show the synthetic procedure (Fig. C11). Solution A: zinc(II) nitrate hexahydrate (Zn(NO₃)₂·6H₂O, 0.84 g, 2.8 mmol, Aladdin, 98.0 %), Fe₂L (0.010 g, 0.015 mmol) and cetyltrimethyl ammonium bromide (CTAB, 0.028 g, 0.077 mmol, J&K, 99.0 %) were dissolved in deionized water (28 mL). Solution B: aqueous solution of 2-methylimidazole (2-MeIm, 12.71 g, 0.15 mol, 196 mL, Acros, 97 %)

Fig. C11. The detailed synthetic path of the macrocyclic-precursor Fe₂L mediated synthesis of Fe₂ DAC.

Q4. All figure captions should be carefully checked for mistakes (for example, Fig. S9 claims that the presented results are for IAV), typos, and signatures without sufficient explanation.

Response:

Thank you for pointing out these typos. We have carefully checked the manuscript and corrected them.

Q5. More representative images of Fe₂ DAC-treated IAVs should be enclosed in supplementary materials in addition to Fig. 4b with a single example.

Response:

Thank you for the suggestion. We have added more images in the Supplementary Figure 9. The lipid envelope of Fe₂ DAC-treated IAV was considerably damaged as shown in Fig. C3.

Fig. C3. TEM images of H1N1 IAVs treated with variable concentrations of Fe₂ DAC.

Q6. All the abbreviations should be explained when first mentioned.

Response:

We have checked and corrected all the abbreviations.

Q7. “Non-specific antiviral effect and application in air filter” section should be rewritten in a clearer manner providing interpretation of any observation. LOX low antiviral activity explanation should be expanded and supplemented with relevant refs.

Response:

We thank the reviewer for the suggestion. We have rewritten this part accordingly. “As a proof-of-concept, we coated different amounts of Fe₂ DAC onto the replacement filter of air purifier for in-situ viral disinfection. As shown in Fig. 5d and Supplementary Fig. 16a, the HA and TCID₅₀ titers were 3.33±0.47 and 3.57±0.01 respectively when having H1N1 virus sprayed on the air filter for 90 min. In comparison, the incorporation of Fe₂ DAC onto the filter significantly reduced both the HA and TCID₅₀ titers of H1N1 virus. In specific, with IAVs sprayed onto the outermost layer, Fe₂ DAC at a concentration of 1 mg cm⁻² reduced both the HA and TCID₅₀ titers of the virus to zero after a two-hour incubation under ambient conditions. In addition, the antiviral effect of Fe₂ DAC was not affected whether it was coated on nonwoven or gauze (Supplementary Fig. 16).”

We acknowledge that understanding LOX low antiviral activity is very important given its high lipid peroxidation activity. We have expanded this part with relevant refs in the revised manuscript. “The low antiviral performance of natural LOX may be caused by different reasons. One plausible explanation is the inactivation of natural LOX caused by the attack of free radicals generated during lipid peroxidation (Supplementary Fig. 11). Another possibility is the poor interaction between LOX and the viral envelop. LOX catalyze the peroxidation of not only free fatty acids but also other complex substrates like intact bio-membranes.^{44,45} But the oxygenation rate of the latter is much lower than that of the free substrates.⁴⁵ Similar to bio-membranes, viral envelope consists of highly packed lipid bilayer which has limited degrees of freedom, leading to the unsatisfactory antiviral performance.”

Q8. Figures quality should be increased.

Response:

We are sorry about the low Figure quality. It is probably due to unexpected file compression during the PDF merging step in the manuscript submission system. We have re-imported the original figures.

Reviewer #3 (Remarks to the Author):

The manuscript "Diatomic Iron Nanozyme with Lipoxidase-like Activity for Efficient Inactivation of Enveloped Virus" by Li and coworkers presents an experimental and computational study of Fe₂ doped dual atom catalyst on graphitic surface as potential nano-isozyme like activity. Increasing the number of metal dopants at the active site has been shown to increase the efficiency of the catalysts and enhance the activity for the specific enzymatic mechanism. The manuscript has been well written. To characterize the structural morphology of the catalyst, the authors have performed X-ray spectroscopic studies combined with theoretical calculations. Following the DFT calculations, a mechanistic pathway has been proposed.

However, I am not convinced about the conclusions drawn from the theoretical observations. In addition, the reaction mechanism proposed in this study has a few drawbacks. Therefore, before having the final decision from my side, I would like to suggest the authors comment on the following points.

We thank the reviewer for the thoughtful and detailed comments. We acknowledge that the proposed mechanism should be provided with more experimental and theoretical support.

Q1. The authors have characterized a four-step mechanism. The reaction proceeds with the initial TS of 0.96 eV. I believe this value would be higher for the reaction to proceed with greater yield. This is not in agreement with the experimental results, where it shows that the diatom catalyst has the better performance as the LOX substrate. Either the reaction proceeds with the alternate pathway, or the some of the environmental interactions are not taken care of, such as the solvent interactions. The authors should comment if the reaction conditions are taken care of for the calculations.

Response:

We are thankful to the referee for this excellent comment which is added to our revised manuscript. As pointed out by the reviewer, the LOX-like activity proceeds with the initial TS of 0.96 eV in the gas phase, and the solvent interactions have not been considered in the DFT calculations. The solvent effects have been added in our revised DFT calculations. The LOX-like activity of Fe₂ DAC nanozyme occurs in an aqueous phase in our experiments, and in order to better describe the catalytic mechanism, the dielectric constant (EB_K=80) of aqueous solvents is considered to model the water environment. As shown in **Figure 6**, the revised calculations indicate that the value of initial TS is reduced to 0.54 eV, taking into account the water environment.

Q2. The authors have proposed that the diene substrate is adsorbed first and then the O₂. The authors should point out what led them to propose this mechanism. Is there any alternative mechanism possible for this process? The authors have not elucidated the different possible reaction mechanism. It might be that O₂ molecule is activated by the diatom dopant, which reacts with the diene to give the respective product.

Response:

We are thankful to the referee for this comment, which gives us an opportunity to elucidate the possible reaction mechanism in LOX-like catalytic activity of Fe₂ DAC nanozyme. In this manuscript, the four-step mechanism used in our calculations is based on the catalytic cycle corresponding to the dioxygenation of a substrate containing a *cis,cis*-1,4-pentadiene moiety by natural LOX enzymes. (Biochimie 178, 170-180(2020); Chem. Rev. 111, 5866-5898 (2011)) The four-step mechanism containing (1) substrate adsorption described by eq 2 corresponding to state 1 to state 2; (2) hydrogen abstraction and radical re-arrangement described by eq 3 corresponding to state 2 to state 3; (3) peroxy radical insertion described by eq 4 corresponding to state 3 to state 4; and (4) product desorption described by eq 5 corresponding to state 4 to state 1.

As for the other possible reaction mechanism mentioned by the reviewer that the O₂ molecule is activated by the diatom dopant, which has been verified by both experimental evidence and theoretical calculations. The experimental evidence show that the OOH free radicals can be detected in the presence of both liposome and O₂, while there is no OOH free radicals only in the presence of O₂ as shown in Fig. C12a, which indicates that the only O₂ cannot be activated to generate OOH free radicals. And the DFT calculations indicate that there are higher energy barriers in four-step mechanism of the O₂ activation by the diatom sites compared to that of the diene activated by the diatom dopant. Taken together, the experimental results and DFT calculations do not support the mechanism that the O₂ molecule is activated by Fe₂ and reacts with the diene to achieve LOX-like activity.

Fig. C12. a, OOH free radicals detection. b, The proposed reaction pathway of LOX-like activity, involving the dioxygenation of a substrate containing a *cis,cis*-1,4-pentadiene moiety on Fe₂ DAC model. Bond distance unit: Å. c, Gibbs free energy profile for key intermediate and transition states in the LOX-like catalytic cycle. Energy unit: eV.

Reviewers' Comments:

Reviewer #1:

Remarks to the Author:

The authors addressed all the points raised in the first review round. The additional results strengthen the manuscript and I consider the manuscript ready for publication.

Reviewer #2:

Remarks to the Author:

The overall quality of the article increased significantly after corrections made by the authors, however the following comments should be considered prior to publication:

- 1) All the details of how this statistics has been calculated should be added to Supplementary Information; moreover, some of these details (how many sample counts were taken for each percentage calculation and how many repetitions were made) should be added to figure description.
- 2) The question Q5 remains unresolved, since this mechanism should be added to the main text; what is more important, the mechanism should explain the differences in results for HA, NA, and NP - therefore, it should be further improved.
- 3) On Fig. C9 mean gray value units should be either converted to more interpretable units or normalized per control to make it dimensionless.
- 4) Additional question Q1 remains unresolved, and characterization section in SI still needs structurization. Reviewer recommends to make separate paragraphs depending on the experiment it connects with; otherwise it is hard to navigate this section.
- 5) Additional question Q3 needs work, since the synthesis steps should include all formulas and compound transformations, othewise it is hard to say what bonds are formed during these steps, and the mechanism of these reactions is not clear.

Additionally, all the data given in the response only should be added either to main or supplementary text.

Reviewer #3:

Remarks to the Author:

The authors have addressed the comments raised earlier. The article is suitable for publication.

Thank you very much.

REVIEWER COMMENTS

Reviewer #2 (Remarks to the Author):

The overall quality of the article increased significantly after corrections made by the authors, however the following comments should be considered prior to publication:

Q1. All the details of how this statistics has been calculated should be added to Supplementary Information; moreover, some of these details (how many sample counts were taken for each percentage calculation and how many repetitions were made) should be added to figure description.

Response:

Thank you very much for the suggestions. The statistics of spherical liposomes were calculated by Image J software, which has been added in the revised supplementary information. The sample counts are more than 100, and all the experiments have been repeated for three times independently. These have been added to figure description (Supplementary Fig. 8). Furthermore, the raw data of Fig.C1 are listed in the following tables. All the source data have been uploaded to figshare.

Fig. C1. Destruction of liposomes			
	Total amount of liposomes	Amount of broken liposomes	Percentage of broken liposomes (%)
Liposome	304	56	18.4%
	393	118	30.0%
	194	51	26.3%
Fe ₂ DAC-treated	541	396	73.2%
	291	181	62.2%
	1424	1023	71.8%
FeSA-treated	666	212	31.8%
	321	120	37.4%
	650	220	33.8%
LOX-treated	147	93	63.3%
	173	100	57.8%
	129	67	51.9%

Supplementary Fig. 8. Destruction of liposomes. a, SEM images and b, quantitative statistics of liposomes treated by Fe₂ DAC, Fe SAC and LOX (n ≥ 100). All the experiments have been repeated for three times independently and the error bars represent standard deviations for three independent measurements.

Q2. The question Q5 remains unresolved, since this mechanism should be added to the main text; what is more important, the mechanism should explain the differences in results for HA, NA, and NP - therefore, it should be further improved.

“Q5. Authors should make more emphasis on viral proteins disruption by Fe₂ DAC including proposed mechanism and quantify these results.”

Response:

Thanks for pointing out this issue. We have re-conducted the WB and confirmed the disruption of proteins with Fe₂ DAC treatment. The mechanism of protein destruction has been added to the main text in the revised manuscript.

“As shown in Fig. 4d,e, HA, NA, and NP proteins were disrupted in a dose-dependent manner by Fe₂ DAC. Treatment with 500 µg mL⁻¹ Fe₂ DAC resulted in undetectable levels of HA and NA proteins, indicating their complete degradation, which is reasonable due to their close association with the lipid envelope. While the destructive effect of Fe₂ DAC on NP protein is weaker than that of HA and NA, which may be because NP protein is spatially away from the envelope. To exclude the interference of Fe₂ DAC to the western blot (WB) technique, relevant controlled trials were also performed (Supplementary Fig.10). Previous studies have suggested that the disruption of the viral proteins could be attributed to the attack of free radicals generated by lipid peroxidation.^{42,43} Here, a similar mechanism was confirmed by electron paramagnetic resonance (EPR) spectrometry, which revealed that peroxy radical, superoxide radical, and hydroxyl radical could be detected only in the presence of Fe₂ DAC and liposome (Supplementary Fig.11).”

Q3. On Fig. C9 mean gray value units should be either converted to more interpretable units or normalized per control to make it dimensionless.

Response:

Thanks for this valuable suggestion. To make mean gray value units dimensionless, we have normalized per control in the Figure.

Q4 Additional question Q1 remains unresolved, and characterization section in SI still needs structurization. Reviewer recommends to make separate paragraphs depending on the experiment it connects with; otherwise it is hard to navigate this section.

“Q1. Characterization section in SI should be further structured.”

Response:

We have refined the characterization section into four parts, including: a) electron microscopy; b) X-ray based structural characterization; c) Single crystal analysis of Fe₂L complex; and d) EPR test. Please refer to the Supplementary Information for details.

Q5. Additional question Q3 needs work, since the synthesis steps should include all formulas and compound transformations, otherwise it is hard to say what bonds are formed during these steps, and the mechanism of these reactions is not clear.

“Q3. All the synthesis steps should be visualized in more details.”

Response:

Thanks for the suggestion. We have drawn the synthetic chemical equations of Fe₂L precursor in the revised Supplementary Information. The synthesis of Fe₂ DAC involves high-temperature pyrolysis, during which the Fe₂L precursor encapsulated in ZIF-8 carbonized. The chemical transformation in this step is complicated and the mechanism is unfortunately unknown.

The synthetic chemical equations of the macrocyclic-precursor Fe₂L complex.

Additionally, all the data given in the response only should be added either to main or supplementary text.

Response:

All the data given in the response have been added to revised manuscript and the revised supplementary information.

Reviewers' Comments:

Reviewer #2:

Remarks to the Author:

All necessary corrections for publication were made by the authors.